# Universal Off-Policy Evaluation

**Yash Chandak**
University of Massachusetts

**Scott Niekum**
University of Texas Austin

**Bruno Castro da Silva**
University of Massachusetts

**Erik Learned-Miller**
University of Massachusetts

**Emma Brunskill**
Stanford University

**Philip S. Thomas**
University of Massachusetts

## Abstract

When faced with sequential decision-making problems, it is often useful to be able to predict what would happen if decisions were made using a new policy. Those predictions must often be based on data collected under some previously used decision-making rule. Many previous methods enable such *off-policy* (or counterfactual) estimation of the *expected* value of a performance measure called the *return*. In this paper, we take the first steps towards a *universal off-policy estimator* (UnO)—one that provides off-policy estimates and high-confidence bounds for *any* parameter of the return distribution. We use UnO for estimating and simultaneously bounding the mean, variance, quantiles/median, inter-quantile range, CVaR, and the entire cumulative distribution of returns. Finally, we also discuss UnO's applicability in various settings, including fully observable, partially observable (i.e., with unobserved confounders), Markovian, non-Markovian, stationary, smoothly non-stationary, and discrete distribution shifts.

## 1 Introduction

Problems requiring sequential decision-making are ubiquitous [5]. When online experimentation is costly or dangerous, it is essential to conduct off-policy evaluation before deploying a new policy; that is, one must leverage existing data collected using some policy $\beta$ (called a behavior policy) to evaluate a performance metric of another policy $\pi$ (called the evaluation policy). For problems with high stakes, such as in terms of health [56] or financial assets [86], it is also crucial to provide high-confidence bounds on the desired performance metric to ensure reliability and safety.

Perhaps the most widely studied performance metric in the off-policy setting is the expected return [83]. However, this metric can be limiting for many problems of interest. Safety-critical applications, such as automated healthcare, require minimizing the chances of risk-prone outcomes, and so performance metrics such as value at risk (VaR) or conditional value at risk (CVaR) are more appropriate [49, 14]. By contrast, applications like online recommendations are subject to noisy data and call for robust metrics like the median and other quantiles [2]. In order to improve user experiences, applications involving direct human-machine interaction, such as robotics and autonomous driving, focus on minimizing uncertainty in their outcomes and thus use metrics like variance and entropy [52, 84]. Recent work in distributional reinforcement learning (RL) have also investigated estimating the cumulative distribution of returns [7, 24] and its various statistical functionals [76]. While it may even be beneficial to use all of these different metrics simultaneously to inform better decision-making, even individually estimating and bounding any performance metric, other than mean and variance, in the *off-policy setting* has remained an open problem.

This raises the main question of interest: *How do we develop a universal off-policy method—one that can estimate any desired performance metrics and can also provide finite-sample confidence bounds that hold simultaneously with high probability for those metrics?*

35th Conference on Neural Information Processing Systems (NeurIPS 2021).

**Prior Work:** Off-policy methods can be broadly categorized as model-based or model-free [83]. Model-based methods typically require strong assumptions on the parametric model when statistical guarantees are needed. Further, using model-based approaches to estimate parameters other than the mean can also require estimating the *distribution* of rewards for *every* state-action pair in order to obtain the complete return distribution for any policy.

By contrast, model-free methods are applicable to a wider variety of settings. Unfortunately, the popular technique of using *importance-weighted returns* [71] only corrects for the *mean* under the off-policy distribution. Recent work by Chandak et al. [18] provides a specialized extension to only correct for the variance. Outside RL, works in the econometrics and causal inference literature have also considered quantile treatments [29, 99] and inferences on counterfactual distributions [28, 20, 36], but these methods are not developed for sequential decisions and do not provide any high-confidence bounds with guaranteed coverage. Further, they often mandate stationarity, identically distributed data, and full observability (i.e., no confounding).

Existing frequentist high-confidence bounds are not only specifically designed for either the mean or variance, but also hold only *individually* [92, 45, 18]. Instead of frequentist intervals, a Bayesian posterior distribution over the mean return and various statistics of that distribution can also be obtained [105]. We are not aware of any method that provides off-policy bounds or even estimates for *any* parameter of the return, while also handling different domain settings that are crucial for RL related tasks. Therefore, a detailed discussion of existing work is deferred to Appendix C.

**Contributions:** We take the first steps towards a *un*iversal *o*ff-policy estimator (UnO) that estimates and bounds the *entire distribution* of returns, and then derives estimates and simultaneous bounds for all parameters of interest. With UnO, we make the following contributions:

**A.** For *any* distributional parameter (mean, variance, quantiles, entropy, CVaR, CDF, etc.), we provide an off-policy method to obtain **(A.1)** model-free estimators; **(A.2)** high-confidence bounds that have guaranteed coverage *simultaneously* for all parameters and that, perhaps surprisingly, often nearly match or outperform prior bounds specifically designed for the mean and the variance; and **(A.3)** approximate bounds using statistical bootstrapping that can often be significantly tighter.

**B.** The above advantages hold for **(B.1)** fully observable and partially observable (i.e., with unobserved confounders) settings, **(B.2)** Markovian and non-Markovian settings, and **(B.3)** settings with stationary, smoothly non-stationary, and discrete distribution shifts in a policy's performance.

**Limitations:** Our method uses importance sampling and thus **(1)** Requires knowledge of action probabilities under the behavior policy $\beta$, **(2)** Any outcome under the evaluation policy should have a sufficient probability of occurring under $\beta$, and **(3)** Variance of our estimators scales exponentially with the horizon length [39, 57], which may be unavoidable in non-Markovian domains [46].

**Notation:** For brevity, we first restrict our focus to the stationary setting. In Section 5, we discuss how to tackle non-stationarity and distribution shifts. A *partially observable Markov decision process* (POMDP) is a tuple $(\mathcal{S}, \mathcal{O}, \mathcal{A}, \mathcal{P}, \Omega, \mathcal{R}, \gamma, d_0)$, where $\mathcal{S}$ is the set of states, $\mathcal{O}$ is the set of observations, $\mathcal{A}$ is the set of actions, $\mathcal{P}$ is the transition function, $\Omega$ is the observation function, $\mathcal{R}$ is the reward function, $\gamma \in [0, 1]$ is the discount factor, and $d_0$ is the starting state distribution. Although our results extend to the continuous setting, for notational ease, we consider $\mathcal{S}, \mathcal{A}, \mathcal{O}$, and the set of rewards to be finite. Since the true underlying states are only partially observable, the resulting rewards and transitions from one partially observed state to another are therefore also potentially non-Markovian [80]. We write $S_t, O_t, A_t$, and $R_t$ to denote random variables for state, observation, action, and reward respectively at time $t$. Let $\mathcal{D}$ be a data set $(H_i)_{i=1}^n$ collected using *behavior policies* $(\beta_i)_{i=1}^n$, where each $H_i$ denotes the *observed trajectory* $(O_0, A_0, \beta(A_0|O_0), R_0, O_1, ...)$. Notice that an observed trajectory contains $\beta(A_t|O_t)$ and does not contain the states $S_t$, for all $t$. Let $G_i := \sum_{j=0}^T \gamma^j R_j$ be the *return* of $H_i$, where $\forall i,\ G_{\min} < G_i < G_{\max}$ for some finite constants $G_{\min}$ and $G_{\max}$, and $T$ is a finite horizon length. Let $G_\pi$ and $H_\pi$ be the random variables for returns and complete trajectories under any policy $\pi$, respectively. Since the set of observations, actions, and rewards are finite, and $T$ is finite, the total number of possible trajectories is finite. Let $\mathcal{X}$ be the finite set of returns corresponding to these trajectories. Let $\mathscr{H}_\pi$ be the set of all possible trajectories for any policy $\pi$. Sometimes, to make the dependence explicit, we write $g(h)$ to denote the return of trajectory $h$. Further, to ensure that samples in $\mathcal{D}$ are informative, we make a standard assumption that any outcome under $\pi$ has sufficient probability of occurring under $\beta$ (see Appendix B.1 for further discussion of assumptions in general),

**Assumption 1.** *The set $\mathcal{D}$ contains independent (not necessarily identically distributed) observed trajectories generated using $(\beta_i)_{i=1}^n$, such that for some (unknown) $\varepsilon > 0$, $(\beta_i(a|o) < \varepsilon) \implies (\pi(a|o) = 0)$, for all $o \in \mathcal{O}, a \in \mathcal{A}$, and $i \in \{1, 2, ..., n\}$.*

## 2 Idea Summary

For the desired universal method, instead of considering each parameter individually, we suggest estimating the entire *cumulative distribution function* (CDF) of returns first:

$$\forall \nu \in \mathbb{R}, \qquad F_\pi(\nu) := \Pr\Big(G_\pi \leq \nu\Big).$$

Any distributional parameter, $\psi(F_\pi)$, can then be estimated from the estimate of $F_\pi$. However, we only have off-policy data from a behavior policy $\beta$, and the typical use of importance sampling [71] only corrects for the mean return. To overcome this, we propose an estimator $\hat{F}_n$ that uses importance sampling from the *perspective of the CDF* to correct for the *entire* distribution of returns. The CDF estimate, $\hat{F}_n$, is then used to obtain a plug-in estimator $\psi(\hat{F}_n)$ for any distributional parameter $\psi(F_\pi)$.

Next, we show that this CDF-centric perspective provides the additional advantage that, if we can compute a $1 - \delta$ *confidence band* $\mathcal{F} : \mathbb{R} \to 2^{\mathbb{R}}$ such that

$$\Pr\Big(\forall \nu \in \mathbb{R}, \ \Pr\big(G_\pi \leq \nu\big) \in \mathcal{F}(\nu)\Big) \geq 1 - \delta,$$

then a $1 - \delta$ upper (or lower) high-confidence bound on any parameter, $\psi(F_\pi)$, can be obtained by searching for a function $F$ that maximizes (or minimizes) $\psi(F)$ and $\forall \nu \in \mathbb{R}$ has $F(\nu) \in \mathcal{F}(\nu)$.

## 3 UnO: Universal Off-Policy Estimator

In the *on-policy* setting, one approach for estimating any parameter of returns, $G_\pi$, might be to first estimate its *cumulative distribution* $F_\pi$ and then use that to estimate its parameter $\psi(F_\pi)$. However, doing this in the off-policy setting requires additional consideration as the *entire* distribution of the observed returns needs to be adjusted to estimate $F_\pi$ since the data is collected using behavior policies that can be different from the evaluation policy $\pi$.

We begin by observing that $\forall \nu \in \mathbb{R}$, $F_\pi(\nu)$ can be expanded using the fact that the probability that the return $G_\pi$ equals $x$ is the sum of the probabilities of the trajectories $H_\pi$ whose return equals $x$,

$$F_\pi(\nu) = \Pr(G_\pi \leq \nu) = \sum_{x \in \mathcal{X}, x \leq \nu} \Pr(G_\pi = x) = \sum_{x \in \mathcal{X}, x \leq \nu} \left( \sum_{h \in \mathscr{H}_\pi} \Pr(H_\pi = h) \mathbb{1}_{\{g(h)=x\}} \right), (1)$$

where $\mathbb{1}_A = 1$ if $A$ is true and 0 otherwise. Now, observing that the indicator function can be one for at most a single value less than $\nu$ as $g(h)$ is a deterministic scalar given $h$, (1) can be expressed as,

$$F_\pi(\nu) = \sum_{h \in \mathscr{H}_\pi} \Pr(H_\pi = h) \sum_{x \in \mathcal{X}, x \leq \nu} \mathbb{1}_{\{g(h)=x\}} = \sum_{h \in \mathscr{H}_\pi} \Pr(H_\pi = h)\Big(\mathbb{1}_{\{g(h)\leq\nu\}}\Big),$$

where the red color is used to highlight changes. Now, from Assumption 1 as $\forall \beta$, $\mathscr{H}_\pi \subseteq \mathscr{H}_\beta$,[1]

$$F_\pi(\nu) = \sum_{h \in \mathscr{H}_\beta} \Pr(H_\pi = h)\Big(\mathbb{1}_{\{g(h)\leq\nu\}}\Big) = \sum_{h \in \mathscr{H}_\beta} \Pr(H_\beta = h)\frac{\Pr(H_\pi = h)}{\Pr(H_\beta = h)}\Big(\mathbb{1}_{\{g(h)\leq\nu\}}\Big). (2)$$

The form of $F_\pi(\nu)$ in (2) is beneficial as it suggests a way to not only perform off-policy corrections for one specific parameter, as in prior works [71, 18], but for the *entire cumulative distribution function* (CDF) of return $G_\pi$. Formally, let $\rho_i := \prod_{j=0}^T \frac{\pi(A_j|O_j)}{\beta_i(A_j|O_j)}$ denote the importance ratio for $H_i$, which is equal to $\Pr(H_\pi = h)/\Pr(H_\beta = h)$ (see Appendix D).

Then, based on (2), we propose the following non-parametric and model-free estimator for $F_\pi$.

$$\forall \nu \in \mathbb{R}, \quad \hat{F}_n(\nu) := \frac{1}{n} \sum_{i=1}^n \rho_i \mathbb{1}_{\{G_i \leq \nu\}}. \tag{3}$$

---

[1]Results can be extended to hybrid probability measures using Radon-Nikodym derivatives.

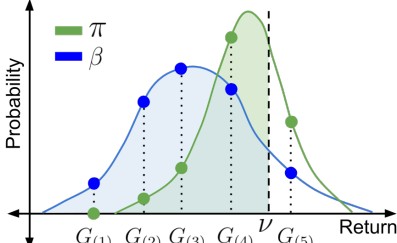

Figure 1: An illustration of return distributions for $\pi$ and $\beta$. The CDF at any point $\nu$ corresponds to the area under the probability distribution up until $\nu$. Having order statistics $(G_{(i)})_{i=1}^5$ of samples $(G_i)_{i=1}^5$ drawn using $\beta$, (3) constructs an empirical estimate of the CDF for $\pi$ (*green* shaded region) by correcting for the probability of observing each $G_i$ using the *importance-sampled counts* of $G_i \leq \nu$. Additionally, weighted-IS (WIS) can be used as in (27) for a variance-reduced estimator for $F_\pi$.

Figure 1 provides intuition for (3). In the following theorem, we establish that this estimator, $\hat{F}_n$, is unbiased and not only pointwise consistent, but also a uniformly consistent estimator of $F_\pi$, even when the data $\mathcal{D}$ is collected using multiple behavior policies $(\beta_i)_{i=1}^n$. The proof (deferred to Appendix D) also illustrates that by using knowledge of action probabilities under the behavior policies, no additional adjustments (e.g., front-door or backdoor [70]) are required by $\hat{F}_n$ to estimate $F_\pi$, even when the domain is non-Markovian or has partial observability (confounders).

**Theorem 1.** *Under Assumption 1, $\hat{F}_n$ is an unbiased and uniformly consistent estimator of $F_\pi$,*

$$\forall \nu \in \mathbb{R}, \quad \mathbb{E}_{\mathcal{D}}\left[\hat{F}_n(\nu)\right] = F_\pi(\nu), \qquad \sup_{\nu \in \mathbb{R}} \left|\hat{F}_n(\nu) - F_\pi(\nu)\right| \xrightarrow{a.s.} 0.$$

**Remark 1.** *Notice that the value of $\hat{F}_n(\nu)$ can be more than one, even though $F_\pi(\nu)$ cannot have a value greater than one for any $\nu \in \mathbb{R}$. This is an expected property of estimators based on importance sampling (IS). For example, the IS estimates of expected return during off-policy mean estimation can be smaller or larger than the smallest and largest possible return when $\rho > 1$.*

Having an estimator $\hat{F}_n$ of $F_\pi$, any parameter $\psi(F_\pi)$ can now be estimated using $\psi(\hat{F}_n)$. However, some parameters like the mean $\mu_\pi$, variance $\sigma_\pi^2$, and entropy $\mathcal{H}_\pi$, are naturally defined using the probability distribution $dF_\pi$ instead of the cumulative distribution $F_\pi$. Similarly, parameters like the $\alpha$-quantile $Q_\pi^\alpha$ and inter-quantile range (which provide tail-robust measures for the mean and deviation from the mean) and conditional value at risk $\text{CVaR}_\pi^\alpha$ (which is a tail-sensitive measure) are defined using the inverse CDF $F_\pi^{-1}(\alpha)$. Therefore, let $(G_{(i)})_{i=1}^n$ be the *order statistics* for samples $(G_i)_{i=1}^n$ and $G_{(0)} := G_{\min}$. Then, we define the off-policy estimator of the inverse CDF for all $\alpha \in [0,1]$, and the probability distribution estimator $d\hat{F}_n$ as,

$$\hat{F}_n^{-1}(\alpha) := \min\left\{g \in (G_{(i)})_{i=1}^n \Big| \hat{F}_n(g) \geq \alpha\right\}, \qquad d\hat{F}_n(G_{(i)}) := \hat{F}_n(G_{(i)}) - \hat{F}_n(G_{(i-1)}), \quad (4)$$

where $d\hat{F}_n(\nu) := 0$ if $\nu \neq G_{(i)}$ for any $i \in (1, \ldots, n)$. Using (4), we now define off-policy estimators for parameters like the mean, variance, quantiles, and CVaR (see Appendix E.1 for more details on these). This procedure can be generalized to any other parameter of $F_\pi$ for which a sample estimator $\psi(\hat{F}_n)$ can be directly created using $\hat{F}_n$ as a plug-in estimator for $F_\pi$.

$$\mu_\pi(\hat{F}_n) := \sum_{i=1}^n d\hat{F}_n(G_{(i)})G_{(i)}, \qquad \sigma_\pi^2(\hat{F}_n) := \sum_{i=1}^n d\hat{F}_n(G_{(i)})\left(G_{(i)} - \mu_\pi(\hat{F}_n)\right)^2,$$

$$Q_\pi^\alpha(\hat{F}_n) := \hat{F}_n^{-1}(\alpha), \qquad \text{CVaR}_\pi^\alpha(\hat{F}_n) := \frac{1}{\alpha}\sum_{i=1}^n d\hat{F}_n(G_{(i)})G_{(i)}\mathbb{1}_{\left\{G_{(i)} \leq Q_\pi^\alpha(\hat{F}_n)\right\}}.$$

**Remark 2.** *Let $H_i$ be the observed trajectory for the $G_i$ that gets mapped to $G_{(i)}$ when computing the order statistics. Note that $d\hat{F}_n(G_{(i)})$ equals $\rho_i/n$ for this $H_i$. This implies that the estimator for the mean, $\mu_\pi(\hat{F}_n)$, reduces exactly to the existing full-trajectory-based IS estimator [71].*

Notice that many parameters and their sample estimates discussed above are nonlinear in $F_\pi$ and $\hat{F}_n$, respectively (the mean is one exception). Therefore, even though $\hat{F}_n$ is an unbiased estimator of $F_\pi$, the sample estimator, $\psi(\hat{F}_n)$, may be a biased estimator of $\psi(F_\pi)$. This is expected behavior because even in the on-policy setting it is not possible to get unbiased estimates of some parameters (e.g., standard deviation), and UnO reduces to the on-policy setting when $\pi = \beta$. However, perhaps surprisingly, we establish in the following section that even when $\psi(\hat{F}_n)$ is a biased estimator of $\psi(F_\pi)$, high-confidence upper and lower bounds can still be computed for both $F_\pi$ and $\psi(F_\pi)$.

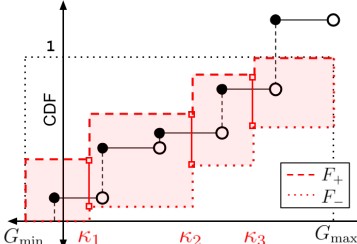

Figure 2: An illustration of $\hat{F}_n$ (in black) using five return samples and the confidence band $\mathcal{F}$ (red shaded region) computed using (5) with confidence intervals (red lines) at three key points $(\kappa_i)_{i=1}^3$. Notice that the vertical "steps" in $\hat{F}_n$ can be of different heights and their total can be greater than 1 due to importance weighting. However, since we know that $F_\pi$ is never greater than 1, $\mathcal{F}$ can be clipped at 1.

## 4   High-Confidence Bounds for UnO

Off-policy estimators are typically prone to high variance, and when the domain can be non-Markovian, the curse of horizon might be unavoidable [46]. For critical applications, this might be troublesome [94] and thus necessitates obtaining confidence intervals to determine how much our estimates can be trusted. Therefore, in this section, we aim to construct a set of possible CDFs $\mathcal{F} : \mathbb{R} \to 2^\mathbb{R}$, called a *confidence band*, such that the true $F_\pi(\nu)$ is within the set $\mathcal{F}(\nu)$ with high probability, i.e., $\Pr(\forall \nu \in \mathbb{R},\ F_\pi(\nu) \in \mathcal{F}(\nu)) \geq 1 - \delta$, for any $\delta \in (0, 1]$. Subsequently, we develop finite-sample bounds for any parameter $\psi(F_\pi)$ using $\mathcal{F}$.

In the on-policy setting, $\mathcal{F}$ can be constructed using the DKW inequality [31] and its tight constants [60]. However, its applicability to the off-policy setting is unclear as **(a)** unlike the on-policy CDF estimate, the "steps" of an off-policy CDF estimate are not of equal heights, **(b)** the "steps" do not sum to one (see Figure 2) and the maximum height of the steps need not be known either, and **(c)** DKW assumes samples are identically distributed, however, off-policy data $\mathcal{D}$ might be collected using multiple different behavior policies. This raises the question: *How do we obtain $\mathcal{F}$ in the off-policy setting?*

Before constructing a confidence band $\mathcal{F}$, let us first focus on obtaining bounds for a single point, $F_\pi(\kappa)$. Let $X \coloneqq \rho(\mathbb{1}_{\{G \leq \kappa\}})$. Then, from Theorem 1, we have that $\mathbb{E}_\mathcal{D}[X] = F_\pi(\kappa)$. This implies that a confidence interval for the mean of $X$ provides a confidence interval for $F_\pi(\kappa)$. Using this observation, existing confidence intervals for the mean of a bounded random variable can be directly applied to $X$ to obtain a confidence interval for $F_\pi(\kappa)$. For example, Thomas et al. [91] present tight bounds for the mean of IS-based random variables by mitigating the variance resulting from the heavy tails associated with IS; we use their method on $\hat{F}_n(\kappa)$ to bound $F_\pi(\kappa)$. Alternatively, recent work by Kuzborskij et al. [53] can potentially be used with a WIS-based $F_\pi$ estimate (27).

Before moving further, we introduce some additional notation. Let $(\kappa_i)_{i=1}^K$ be any $K$ "key points" and let $\texttt{CI}_-(\kappa_i, \delta_i)$ and $\texttt{CI}_+(\kappa_i, \delta_i)$ be the lower and the upper confidence bounds on $F_\pi(\kappa_i)$ constructed at each key point using the observation made in the previous paragraph, such that

$$\forall i \in (1, ..., K), \quad \Pr\left(\texttt{CI}_-(\kappa_i, \delta_i) \leq F_\pi(\kappa_i) \leq \texttt{CI}_+(\kappa_i, \delta_i)\right) \geq 1 - \delta_i.$$

We now use the following observation to obtain a band, $\mathcal{F}$, that contains $F_\pi$ with high confidence. Because $F_\pi$ is a CDF, it is necessarily monotonically non-decreasing, and so if $F_\pi(\kappa_i) \geq \texttt{CI}_-(\kappa_i, \delta_i)$ then for any $\nu \geq \kappa_i$, $F_\pi(\nu)$ must be no less than $\texttt{CI}_-(\kappa_i, \delta_i)$. Similarly, if $F_\pi(\kappa_i) \leq \texttt{CI}_+(\kappa_i, \delta_i)$ then for any $\nu \leq \kappa_i$, $F_\pi(\nu)$ must also be no greater than $\texttt{CI}_+(\kappa_i, \delta_i)$. Let $\kappa_0 \coloneqq G_{\min}$, $\kappa_{K+1} \coloneqq G_{\max}$, $\texttt{CI}_-(\kappa_0, \delta_0) \coloneqq 0$, and $\texttt{CI}_+(\kappa_{K+1}, \delta_{K+1}) \coloneqq 1$; then, as illustrated in Figure 2, we can construct a lower function $F_-$ and an upper function $F_+$ that encapsulate $F_\pi$ with high probability,

$$F_-(\nu) \coloneqq \begin{cases} 1 & \text{if } \nu > G_{\max}, \\ \max_{\kappa_i \leq \nu} \texttt{CI}_-(\kappa_i, \delta_i) & \text{otherwise.} \end{cases} \qquad F_+(\nu) \coloneqq \begin{cases} 0 & \text{if } \nu < G_{\min}, \\ \min_{\kappa_i \geq \nu} \texttt{CI}_+(\kappa_i, \delta_i) & \text{otherwise.} \end{cases} \tag{5}$$

**Theorem 2.** *Under Assumption 1, for any $\delta \in (0, 1]$, if $\sum_{i=1}^K \delta_i \leq \delta$, then the confidence band defined by $F_-$ and $F_+$ provides guaranteed coverage for $F_\pi$. That is,*

$$\Pr\left(\forall \nu \in \mathbb{R},\ F_-(\nu) \leq F_\pi(\nu) \leq F_+(\nu)\right) \geq 1 - \delta.$$

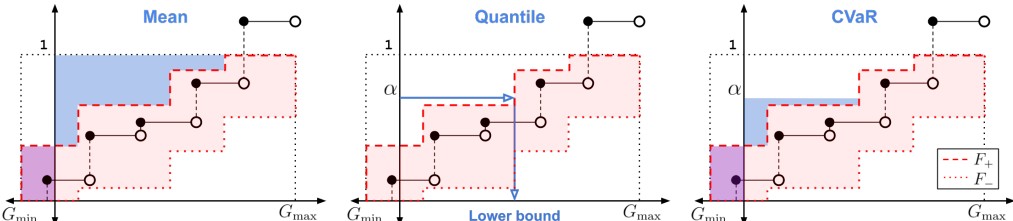

Figure 3: Given a confidence band $\mathcal{F}$, bounds for many parameters can be obtained using geometry. **(Left)** For a lower bound on the mean, we would want a CDF $F \in \mathcal{F}$ that assigns as high a probability as possible on lower $G$ values, and $F_+$ is the CDF which does that. To obtain the mean of $F_+$, we use the property that the mean of a distribution is the area above the CDF on the positive x-axis minus the area below the CDF on the negative x-axis [3]. Hence, the mean of the distribution characterized by $F_+$ is the area of the shaded blue region minus the area of the shaded purple region, and this value is the high-confidence lower bound on the mean. **(Middle)** Similarly, within $\mathcal{F}$, $F_+$ characterizes the distribution with the smallest $\alpha$-quantile. **(Right)** Building upon the lower bounds for the mean and the quantile, Thomas and Learned-Miller [90] showed that the lower bound for $\alpha$-CVaR can be obtained using the area of the shaded blue region minus the area of the shaded purple region, normalized by $\alpha$. To get the upper bounds on the mean, quantile, and CVaR, analogous arguments hold using the lower bound CDF $F_-$. See Appendix E.5 for discussions of variance, inter-quantile, entropy, and other parameters.

**Remark 3.** *Notice that any choice of $(\kappa_i)_{i=1}^{K}$ results in a valid band $\mathcal{F}$. However, $\mathcal{F}$ can be made tighter by optimizing over the choice of $(\kappa_i)_{i=1}^{K}$. In Appendix E.5, we present one such method using cross-validation to minimize the area enclosed within $\mathcal{F}$.*

Having obtained a high-confidence band for $F_\pi$, we now discuss how high-confidence bounds for any parameter $\psi(F_\pi)$ can be obtained using this band. Formally, with a slight overload of notation let $\mathcal{F}$ be the set of all possible CDFs bounded between $F_-$ and $F_+$, that is,

$$\mathcal{F} := \left\{ F \;\middle|\; \forall \nu \in \mathbb{R}, \; F_-(\nu) \le F(\nu) \le F_+(\nu) \right\}.$$

This band $\mathcal{F}$ contains many possible CDFs, one of which is $F_\pi$ with high probability. Therefore, to get a lower or upper bound, $\psi_-$ or $\psi_+$, on $\psi(F_\pi)$, we propose deriving a CDF $F \in \mathcal{F}$ that minimizes or maximizes $\psi(F)$, respectively, and we show that these contain $\psi(F_\pi)$ with high probability:

$$\psi_- := \inf_{F \in \mathcal{F}} \psi(F), \qquad \psi_+ := \sup_{F \in \mathcal{F}} \psi(F). \tag{6}$$

**Theorem 3.** *Under Assumption 1, for any $1 - \delta$ confidence band $\mathcal{F}$, the confidence interval defined by $\psi_-$ and $\psi_+$ provides guaranteed coverage for $\psi(F_\pi)$. That is,*

$$\Pr\left( \psi_- \le \psi(F_\pi) \le \psi_+ \right) \ge 1 - \delta.$$

While obtaining $\psi_-$ might not look straightforward, one can obtain closed-form expressions for many popular parameters of interest. In other cases, simple algorithms exist for computing $\psi_-$ and $\psi_+$ [74]. Figure 3 provides geometric depictions of the closed-form expressions for some parameters.

**Remark 4.** *Perhaps surprisingly, even though $\psi(\hat{F}_n)$ may be biased, we can obtain high-confidence bounds with guaranteed coverage on any $\psi(F_\pi)$ using the confidence band $\mathcal{F}$. In fact, confidence bounds for* all *parameters computed using (6) hold* simultaneously *with probability at least $1 - \delta$ as they are all derived from the same confidence band, $\mathcal{F}$.*

**3.1. Statistical Bootstrapping:** An important advantage of having constructed an off-policy estimator of any $\psi(F_\pi)$ is that it opens up the possibility of using *resampling*-based methods, like statistical bootstrapping [32], to obtain *approximate* confidence intervals for $\psi(F_\pi)$. In particular, we can use the *bias-corrected and accelerated* (BCa) bootstrap procedure to obtain $\psi_-$ and $\psi_+$ for $\psi(F_\pi)$. This procedure is outlined in Algorithm 1 in Appendix E.4.

Unlike the bounds from (6), BCa-based bounds do not offer guaranteed coverage and need to be computed individually for each parameter $\psi$. However, they can be combined with UnO to get significantly tighter bounds with less data, albeit without guaranteed coverage.

# 5   Confounding, Distributional Shifts, and Smooth Non-Stationarities

A particular advantage of UnO is the remarkable simplicity with which the estimates and bounds for $F_\pi$ or $\psi(F_\pi)$ can be extended to account for confounding, distributional shifts, and smooth non-stationarities that are prevalent in real-world applications [30].

**Confounding / Partial Observability:** Estimator $\hat{F}_n$ in (3) accounts for partial observability when both $\pi$ and $\beta$ have the same observation set. However, in systems like automated loan approval [94], data might have been collected using a behavior policy $\beta$ dependent on sensitive attributes like race and gender that may no longer be allowable under modern laws. This can make the available observation, $\widetilde{O}$, for an evaluation policy $\pi$ different from the observations, $O$, for $\beta$, which may also have been a partial observation of the underlying true state $S$.

However, an advantage of many such automated systems (e.g., online recommendation, automated healthcare, robotics) is the direct availability of behavior probabilities $\beta_i(A|O)$. In Appendix D, we provide generalized proofs for all the earlier results, showing that access to $\beta_i(A|O)$ allows UnO to handle various sources of confounding even when $\widetilde{O} \neq O$, without requiring any additional adjustments. When $\beta_i(A|O)$ is not available, we allude to possible alternatives in Appendix B.1.

**Distribution Shifts:** Many practical applications exhibit distribution shifts that might be discrete or abrupt. One example is when a medical treatment developed for one demographic is applied to another [37]. To tackle discrete distributional shifts, let $F_\pi^{(1)}$ and $F_\pi^{(2)}$ denote the CDFs of returns under policy $\pi$ in the first and the second domain, respectively. To make the problem tractable, similar to prior work on characterizing distribution shifts [10], we assume that the Kolmogorov-Smirnov distance between $F_\pi^{(1)}$ and $F_\pi^{(2)}$ is bounded.

**Assumption 2.** *There exists $\epsilon \geq 0$, such that $\sup\limits_{\nu \in \mathbb{R}} \left| F_\pi^{(1)}(\nu) - F_\pi^{(2)}(\nu) \right| \leq \epsilon$.*

Given data $\mathcal{D}$ collected in the first domain, one can obtain the bounds $F_-^{(1)}$ and $F_+^{(1)}$ on $F_\pi^{(1)}$ as in Section 4. Now since $F_\pi^{(2)}$ can differ from $F_\pi^{(1)}$ by at most $\epsilon$ at any point, we propose the following bounds for $F_\pi^{(2)}$ for all $\nu \in \mathbb{R}$ and show that they readily provide guaranteed coverage for $F_\pi^{(2)}$:

$$F_-^{(2)}(\nu) := \max(0, F_-^{(1)}(\nu) - \epsilon), \qquad F_+^{(2)}(\nu) := \min(1, F_+^{(1)}(\nu) + \epsilon). \tag{7}$$

**Theorem 4.** *Under Assumptions 1 and 2, $\forall \delta \in (0, 1]$, the confidence band defined by $F_-^{(2)}$ and $F_+^{(2)}$ provides guaranteed coverage for $F_\pi^{(2)}$. That is, $\Pr(\forall \nu, \ F_-^{(2)}(\nu) \leq F_\pi^{(2)}(\nu) \leq F_+^{(2)}(\nu)) \geq 1 - \delta$.*

**Smooth Non-stationarity:** The stationarity assumption is unreasonable for applications like online tutoring or recommendation systems, which must deal with drifts of students' interests or seasonal fluctuations of customers' interests [93, 88]. In the worst case, however, even a small change in the transition dynamics can result in a large fluctuation of a policy's performance and make the problem intractable. Therefore, similar to the work of Chandak et al. [16], we assume that the distribution of returns for any $\pi$ changes smoothly over the past episodes 1 to $L$, and the $\ell$ episodes in the future. In particular, we assume that the trend of $F_\pi^{(i)}(\nu)$ for all $\nu$ can be modeled using least-squares regression using a nonlinear basis function $\phi : \mathbb{R} \to \mathbb{R}^d$ (e.g., the Fourier basis, which is popular for modeling non-stationary trends [12]).

**Assumption 3.** *For any $\nu$, $\exists w_\nu \in \mathbb{R}^d$, such that, $\forall i \in [1, L + \ell], \quad F_\pi^{(i)}(\nu) = \phi(i)^\top w_\nu$.*

Estimating $F_\pi^{(L+\ell)}$ can now be seen as a time-series forecasting problem. Formally, for any key point $\kappa$, let $\hat{F}_n^{(i)}(\kappa)$ be the estimated CDF using $H_i$ observed in episode $i$. From Theorem 1, we know that $\hat{F}_n^{(i)}(\kappa)$ is an unbiased estimator of $F_\pi^{(i)}(\kappa)$; therefore, $(\hat{F}_n^{(i)}(\kappa))_{i=1}^L$ is an unbiased estimate for the underlying time-varying sequence $(F_\pi^{(i)}(\kappa))_{i=1}^L$. Now, using methods from time-series literature, the trend of $(\hat{F}_n^{(i)}(\kappa))_{i=1}^L$ can be analyzed to forecast $F_\pi^{(L+\ell)}(\kappa)$, along with its CIs. In particular, we propose using *wild bootstrap* [58, 26], which provides *approximate* CIs with finite sample error of $O(L^{-1/2})$ while also handling non-normality and heteroskedasticity, which would occur when dealing with IS-based estimates resulting from different behavior polices [16]. See Appendix E.6 for more details. Finally, using the bounds obtained using wild bootstrap at multiple key points, an entire confidence band can be obtained as discussed in Section 4.

# 6 Empirical Studies

In this section, we provide empirical support for the established theoretical results for the proposed UnO estimator and high-confidence bounds. To do so, we use the following domains: **(1)** An open source implementation [102] of the FDA-approved type-1 diabetes treatment simulator [59], **(2)** A stationary and a non-stationary recommender system domain, and **(3)** A continuous-state Gridworld with partial observability, where data is collected using multiple behavior policies. Detailed description for domains and the procedures for obtaining $\pi$ and $\beta$ are provided in Appendix F.1; code is also publicly available here. In the following, we discuss four primary takeaway results.

**(A) Characteristics of the UnO estimator:** Figure 4 reinforces the universality of UnO. As can be seen, UnO can accurately estimate the entire CDF and a wide range of its parameters: mean, variance, quantile, and CVaR.

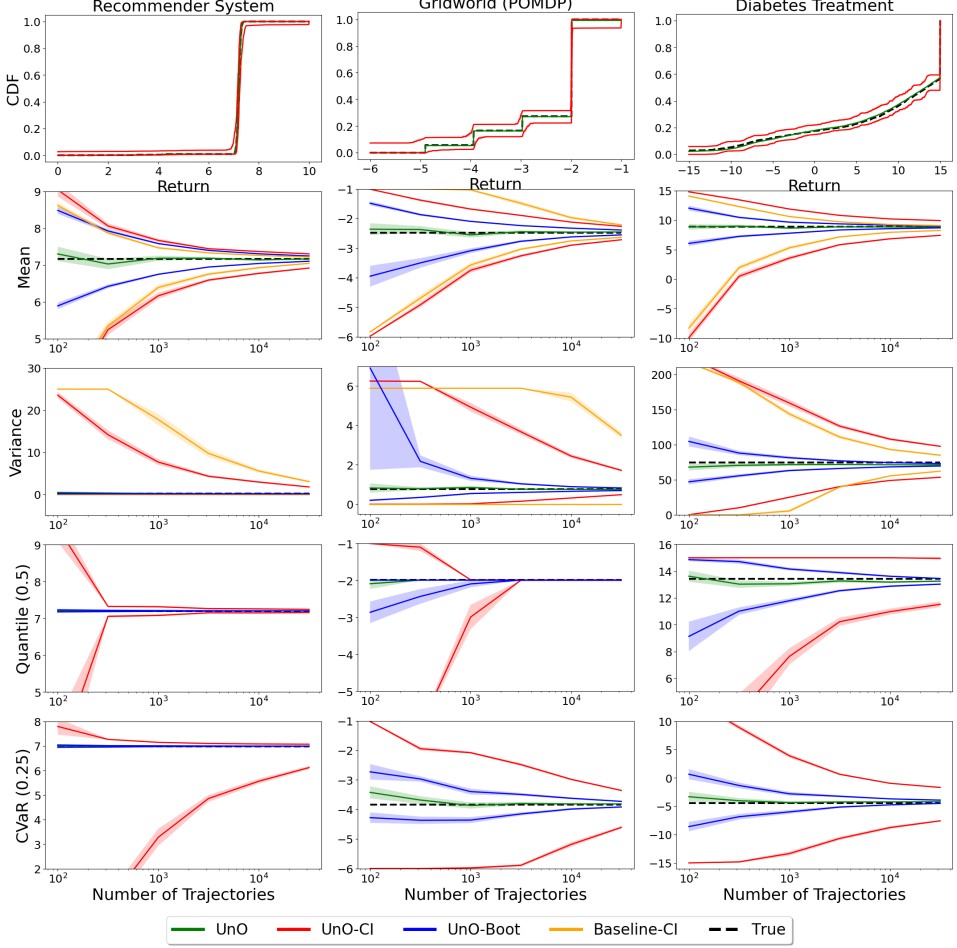

Figure 4: Performance trend of the proposed estimators and bounds on three domains. The black dashed line is the true value of $F_\pi$ or $\psi(F_\pi)$, green is our UnO estimator, red is our `CI`-based UnO bound, blue is the bootstrap version of our UnO bound, and yellow is the baseline bound for the mean [91] or variance [18]. Each bound has two lines (upper and lower); however, some are not visible due to overlaps. The shaded regions are $\pm 2$ standard error, computed using 30 trials. The plots in the top row are for CDFs obtained using $3 \times 10^{4.5}$ samples. The next four rows are for different parameters and share the same x-axis. Bounds were obtained for a failure rate $\delta = 0.05$. Since the UnO-Boot and Baseline-CI methods do not hold simultaneously for all the parameters, they were made to hold with failure rate of $\delta/4$ for a fair comparison (as there are 4 parameters in this plot).

**(B) Comparison of UnO with prior work:** Recent works for bounding the mean [45, 35] assume no confounding and Markovian structure. Therefore, for a fair comparison, we resort to the method

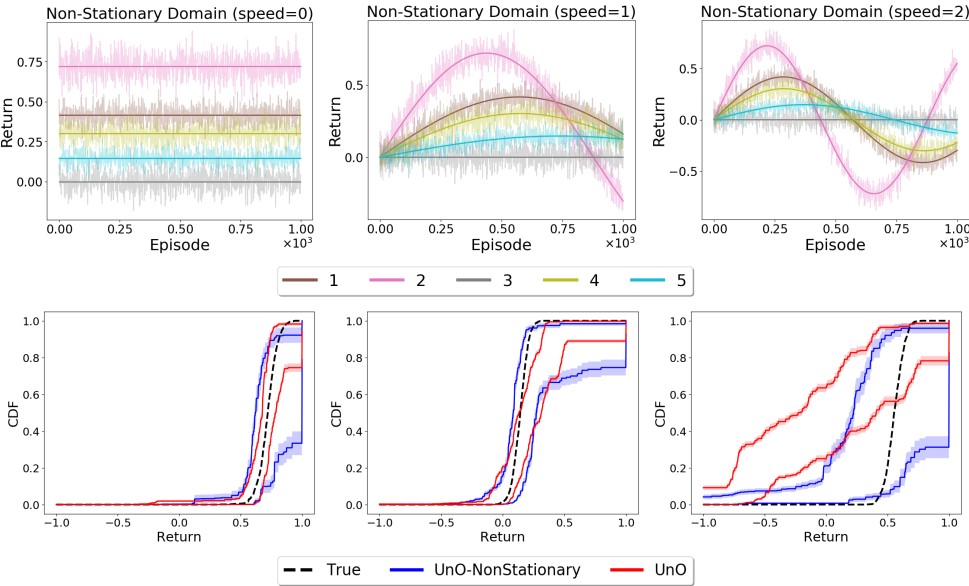

Figure 5: **(Top row)** True rewards (unknown to the RL agent) associated with each of the five items over the past 1000 episodes for different *speeds* of non-stationarity. Speed of 0 indicates stationary setting and higher speeds indicates greater degrees of non-stationarity. **(Bottom row)**. The black dashed line is the true value of the future distribution of returns under $\pi$: $F_\pi^{(L+\ell)}$, where $L = 1000$ and $\ell = 1$. In red is our UnO bound that does not account for non-stationarity, and in blue is the wild-bootstrap version of our UnO bound that accounts for non-stationarity. The shaded region corresponds to one standard error computed using 30 trials. Bounds were obtained for a failure rate $\delta = 0.05$. **(Left column)** In the stationary setting, both the variants of UnO bounds approximately contain the true future CDF $F_\pi^{(L+\ell)}$. In this setting, the UnO method designed only for stationary settings provides a tighter bound. **(Middle & Right columns)** As the domain becomes non-stationary, UnO bounds that do not account for non-stationarity fail to adequately bound the true future CDF $F_\pi^{(L+\ell)}$. When the degree of non-stationarity is high, not accounting for non-stationarity can lead to significantly inaccurate bounds. By comparison, UnO bounds that use wild bootstrap to tackle non-stationarity provide a more accurate bound throughout. As expected, when the fluctuations due to non-stationarity increase, the width of the confidence band increases as well. These results illustrate (a) the importance of accounting for non-stationarity, when applicable, and (b) the flexibility offered by our proposed universal off-policy estimator, UnO, to tackle such settings.

of Thomas et al. [91] that can provide tight bounds even when the domain is non-Markovian or has confounding (partial observability). Perhaps surprisingly, Figure 4 shows that the proposed guaranteed coverage bounds, termed *UnO-CI* here, can be competitive with this existing specialized bound, termed *Baseline-CI* here, for the mean. In fact, UnO-CI can often require an order of magnitude less data compared to the specialized bounds for variance [18]; we refer readers to Appendix F.2 for a discussion on potential reasons. This suggests that the universality of UnO can be beneficial even when only one specific parameter is of interest.

**(C) Finite-sample confidence bounds for other parameters using UnO:** Figure 4 demonstrates that UnO-CI also successfully addresses the open question of providing guaranteed coverage bounds for multiple parameters simultaneously without additional applications of the union bound. As expected, bounds for parameters like variance and CVaR that depend heavily on the distribution tails take more samples to shrink than bounds on other parameters (like the median [quantile(0.5)]). Additional discussion on the observed trends for the bounds is provided in Appendix F.2.

The proposed UnO-Boot bounds, as discussed in Section 3.1, are approximate and might not always hold with the specified probability. However, they stand out by providing *significantly* tighter, and thus more practicable, confidence intervals.

**(D) Results for non-stationary settings:** Results for this setting are presented in Figure 5. As discussed earlier, online recommendation systems for tutorials, movies, advertisements and other

products are ubiquitous. However, the popular assumption of stationarity is seldom applicable to these systems. In particular, personalizing for each user is challenging in such settings as interests of a user for different items among the recommendable products fluctuate over time. For an example, in the context of online shopping, interests of customers can vary based on seasonality or other unknown factors. To abstract such settings, in this domain the reward (interest of the user) associated with each item changes over time. See Figure 5 (top row) for visualization of the domain, for different "speeds" (degrees of non-stationarity).

In all the settings with different speeds, a uniformly random policy was used as a behavior policy $\beta$ to collect data for 1000 episodes. To test the efficacy of UnO, when the future domain can be different from the past domains, the evaluation policy was chosen to be a near-optimal policy for the future episode: $1000 + 1$.

## 7 Conclusion

We have taken the first steps towards developing a universal off-policy estimator (UnO), closing the open question of whether it is possible to estimate and provide finite-sample bounds (that hold with high probability) for *any* parameter of the return distribution in the *off-policy* setting, with minimal assumptions on the domain. Now, without being restricted to the most common and basic parameters, researchers and practitioners can fully characterize the (potentially dangerous or costly) behavior of a policy without having to deploy it.

There are many new questions regarding how UnO can be improved for policy *evaluation* by further reducing data requirements or weakening assumptions. Using UnO for policy *improvement* also remains an interesting future direction. Subsequent to this work, Huang et al. [43] showed how models can be used to obtain UnO-style doubly robust estimators along with its convergence rates in the contextual bandit setting. This allows their method to also provide finite-sample uniform CDF bounds for a broad class of Lipschitz risk functionals.

## 8 Acknowledgements

We thank Shiv Shankar, Scott Jordan, Wes Cowley, and Nan Jiang for the feedback, corrections, and other contributions to this work. We would also like to thank Bo Liu, Akshay Krishnamurthy, Marc Bellemare, Ronald Parr, Josiah Hannah, Sergey Levine, Jared Yeager, and the anonymous reviewers for their feedback on this work.

Research reported in this paper was sponsored in part by a gift from Adobe, NSF award #2018372, and the DEVCOM Army Research Laboratory under Cooperative Agreement W911NF-17-2-0196 (ARL IoBT CRA). Research in this paper is also supported in part by NSF (IIS-1724157, IIS-1638107, IIS-1749204, IIS-1925082), ONR (N00014-18-2243), AFOSR (FA9550-20-1-0077), and ARO (78372-CS, W911NF-19-2-0333). The views and conclusions contained in this document are those of the authors and should not be interpreted as representing the official policies, either expressed or implied, of the Army Research Laboratory or the U.S. Government. The U.S. Government is authorized to reproduce and distribute reprints for Government purposes notwithstanding any copyright notation herein.

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
