# A Notation

| Symbol | Meaning |
|--------|---------|
| $\mathcal{D}$ | Data set of the observed trajectories |
| $n$ | Total number of observed trajectories in $\mathcal{D}$ |
| $\pi$ | Evaluation policy |
| $\beta_i$ | Behavior policy for the $i^{\text{th}}$ trajectory |
| $\rho_i$ | Importance ratio for the observed trajectory $H_i$ |
| $\mathcal{S}$ | State set |
| $\mathcal{O}, \widetilde{\mathcal{O}}$ | Observation set for the behavior policy and the evaluation policy, respectively |
| $\mathcal{A}$ | Action set |
| $\mathcal{P}$ | Transition dynamics, $\mathcal{P} : \mathcal{S} \times \mathcal{A} \to \Delta(\mathcal{S})$ |
| $\mathcal{R}$ | Reward function, $\mathcal{R} : \mathcal{S} \times \mathcal{A} \to \Delta(\mathbb{R})$ |
| $\Omega$ | Observation function for behavior policy, $\Omega : \mathcal{S} \to \Delta(\mathcal{O})$ |
| $\Omega_2$ | Observation function for the evaluation policy, $\Omega_2 : \mathcal{S} \times \mathcal{O} \to \Delta(\widetilde{\mathcal{O}})$ |
| $\gamma$ | Discounting factor |
| $d_0$ | Starting state distribution |
| $T$ | Finite horizon length |
| $H_i, H_\pi$ | $i^{\text{th}}$ observed trajectory in the dataset and complete trajectory under policy $\pi$, respectively |
| $G_i, G_\pi$ | Return observed in the $i^{\text{th}}$ trajectory in the dataset and return under any policy $\pi$, respectively |
| $G_{\min}, G_{\max}$ | Minimum and maximum value of a return, respectively |
| $F_\pi, \mathrm{d}F_\pi$ | True CDF of returns under policy $\pi$ and its associated probability distribution, respectively |
| $\hat{F}_n, \bar{F}_n$ | Off-policy CDF estimator and weighted off-policy CDF estimator using $n$ samples, respectively |
| $F_-, F_+$ | Lower and upper bound on the CDF |
| $\mathcal{F}$ | The set of all CDFs between the upper bound and the lower bounds |
| $\kappa_i, K$ | $i^{\text{th}}$ key point and total number of key points, respectively |
| $\alpha$ | Value for defining inverse CDF-based statistics |
| $\psi$ | Generic functional for a distributional parameter/statistic |
| $\psi_-, \psi_+$ | Lower and upper bounds for $\psi(F_\pi)$ |
| $\delta$ | Failure rate for the bounds |
| $\mathcal{D}_{\text{eval}}, \mathcal{D}_{\text{train}}$ | Evaluation and training split of the dataset $\mathcal{D}$ |
| $\mathtt{CI}_-, \mathtt{CI}_+$ | Lower and upper confidence bounds for a given random variable |
| $\theta$ | Parameters that are used to construct $\mathcal{F}$ |
| $\mathscr{A}$ | Euclidean area enclosed within $\mathcal{F}$ |
| $X_i^*$ | $i^{\text{th}}$ bootstrap resampled value for any random variable $X$ |
| $\varepsilon, \epsilon$ | Some small value in Assumption 1 and Assumption 2, respectively |
| $w_\nu, \phi$ | Regression weights and basis function for the assumption on smooth non-stationarity |
| $L, \ell$ | Number of past and future episodes being considered in the smooth non-stationary setting |

Table 1: List of symbols used in the main paper and their associated meanings.

# B Broader Impact

While our estimators and bounds are both theoretically sound and intuitively simple, it is important for a broader audience to understand the limitations of our method, assumptions being made, and what can be done when these assumptions do not hold. Understanding these assumptions can also help in mitigating any undesired biases in applications built around UnO and can thus avoid any potential negative societal impacts. In the following, we briefly allude to possible alternatives when the required assumptions are violated.

## B.1 Discussion of Assumptions and Requirements of UnO

**Knowledge of Subset Support:** Through Assumption 1, UnO requires that all the behavior policies $(\beta_i)_{i=1}^n$ have sufficient support for actions that have non-zero probability under $\pi$. Particularly, it requires that the $\beta(a|o)$ is bounded below by (an unknown) $\epsilon$ when $\pi(a|o) > 0$. This ensures

that importance ratios are bounded and thus simplifies analysis for UnO's consistency results and constructing confidence intervals. This assumption is common both in the off-policy literature [47, 104, 105] and in real applications [87]

The above assumption is also equivalent to assuming bounded exponentiated-Renyi-divergence (for $\alpha = \infty$) between the probability distributions of trajectories under the behavior and the evaluation policies [61]. As the UnO's bound for the CDF uses CIs for the mean as a sub-routine, the above assumption can be relaxed by using CIs for the mean that depend on Renyi-divergence for other values of $\alpha$ [61]. Similarly, consistency results for UnO rely upon finite variance, which can also be achieved by instead assuming that the Renyi-divergence is bounded for $\alpha = 2$.

Alternatively, Assumption 1 can be relaxed to only absolute continuity by using methods that provide valid CIs for the mean by clipping the importance weights. (See the work by Thomas et al. [91, Theorem 1] for removal of the upper bound on the importance weights when lower-bounding the mean, and the work by Chandak et al. [18, Theorem 5] for removal of the upper bound on the importance weights when upper-bounding the mean). Furthermore, prior work has also shown how even the assumption of absolute continuity can in some cases be removed (See discussion around Eqn 8 in the appendix of the work by Thomas et al. [91]). If the supports for the behavior and the evaluation policies are unequal, Thomas and Brunskill [89] also present a technique to reduce variance resulting from IS.

Further, WIS might also be helpful in relaxing the assumptions on the IS ratios. Specifically, WIS-based mean bounds [53] can also be used along with the WIS-based UnO estimator (27) to get a valid confidence band for the entire CDF.

Using multi-importance sampling (MIS), the subset support requirement for *all* $(\beta_i)_{i=1}^n$ can be relaxed to the requirement that the *union of supports* under the behavior policies $(\beta_i)_{i=1}^n$ has sufficient support [98, 67, 61]. MIS can also help in substantially reducing variance. However, this relaxation requires an alternate assumption that a complete knowledge of all the behavior policies $(\beta_i)_{i=1}^n$, not just the probabilities of the action executed using them, is available.

**Knowledge of Action Probabilities under Behavior Policies** $(\beta_i)_{i=1}^n$**:** UnO requires access to the probability $\beta(a|o)$ (only the scalar probability value and not the entire policy $\beta$) of the actions available in the data set, $\mathcal{D}$, to compute the importance sampling ratios in (3). Access to the probability $\beta(a|o)$ is often available when $\mathcal{D}$ is collected using an automated policy; however, it might not be available in some cases, such as when decisions were previously made by humans.

When the probability $\beta(a|o)$ is not available, one natural alternative is to estimate it from the data and use this estimate of $\beta(a|o)$ in the denominator of the importance ratios. This technique is also known as regression importance sampling (RIS) and is known to provide biased but consistent estimates for the mean [41, 69] in the Markov decision process setting (MDP) setting. For UnO, $\hat{F}_n(\nu)$ is analogous to mean estimation of $X := \rho(\mathbb{1}_{\{G \leq \nu\}})$, for any $\nu$. Therefore, the findings of RIS can be directly extended to UnO in the MDP setting, where $\widetilde{O} = O = S$. In the following, we provide a high-level discussion for the setting when $\beta(a|o)$ is *not* available and the states are partially observed,

- **Partial observability with** $\widetilde{O} = O$**:** In this setting, as $\beta(a|o) = \beta(a|\tilde{o})$, one can use density estimation on the available data, $\mathcal{D}$, to construct an estimator $\hat{\beta}(a|o)$ of $\Pr(a|\tilde{o}) = \beta(a|\tilde{o})$ and use RIS to get a biased but consistent estimator for $F_\pi$. Here, bias results from the estimation error in $\hat{\beta}(a|o)$ but consistency follows as the true $\beta(a|o)$ can be recovered in the limit when $n \to \infty$.

  In context of UnO, using $\hat{\beta}(a|o)$ instead of $\beta(a|o)$ violates the unbiased condition for $\hat{F}_n$, which was necessary to obtain the CIs and construct $\mathcal{F}$. Therefore, high-confidence bounds with guaranteed coverage cannot be obtained using UnO in this setting. However, point estimates and approximate bootstrap bounds can still be obtained.

- **Partial observability with** $\widetilde{O} \neq O$**:** In this setting, using RIS will produce neither an unbiased nor a consistent estimator for $F_\pi$. As $\mathcal{D}$ only has $\tilde{o}$ and not $o$, at best it is only possible to estimate $\Pr(a|\tilde{o}) = \sum_{x \in \mathcal{O}} \beta(a|x) \Pr(x|\tilde{o})$ through density estimation using data $\mathcal{D}$. However, in general, since $\beta(a|o) = \Pr(a|o) \neq \Pr(a|\tilde{o})$ we cannot even consistently estimate the denominator for importance sampling unless some other stronger assumptions are made. See work by Namkoong et al. [65], Tennenholtz et al. [85], Bennett et al. [9] and Kallus and Zhou [48] for possible alternative assumptions and approaches to tackle this setting.

**Knowledge of $G_{\min}, G_{\max}$:** To construct the CDF band $\mathcal{F}$, UnO requires knowledge of $G_{\min}$ and $G_{\max}$ in (5). Notice from Figure 2 that knowing $G_{\max}$ helps in clipping the *lower bound* for the *upper tail* (LBUT) of $\mathcal{F}$, which otherwise would have extended to $+\infty$. Similarly, knowing $G_{\min}$ helps in clipping the *upper bound* for the *lower tail* (UBLT) of $\mathcal{F}$, which otherwise would have extended to $-\infty$.

Typically, even if $G_{\min}$ or $G_{\max}$ is not known, they can be obtained as $R_{\min}/(1-\gamma)$ or $R_{\max}/(1-\gamma)$, respectively, where $R_{\min}$ and $R_{\max}$ are known finite lower and upper bounds for any individual reward. Otherwise, knowledge of $G_{\min}$ or $G_{\max}$ can be relaxed if the desired bound on $\psi$ does not depend on UBLT or LBUT, respectively. For example, observe from Figure 3 that (a) The lower bound for the mean or quantile does not depend on LBUT. Analogously, if only an upper bound for the mean or quantile is required, then UBLT is not needed. (b) The lower bound on CVaR depends on UBLT, however, (for small values of $\alpha$) the upper bound on CVaR neither depends on LBUT nor UBLT. (c) For an upper bound on variance, both LBUT and UBLT are required. However, for the variance's lower bound, neither LBUT nor UBLT are required. See Figure 6 for intuition.

**Knowledge of Function Class $\phi$:** For the smoothly non-stationary setting, through Assumption 3, UnO requires access to the basis functions $\phi$ that can be used with least-squares regression to analyze the trend in the distributions of returns $(F_\pi^{(i)}(\nu))_{i=1}^L$ for any $\nu \in \mathbb{R}$. In practice, one can use sufficiently flexible basis functions to model time-series trends (e.g., Fourier basis [12]). To avoid overfitting or underfitting, one could also use goodness-of-fit tests to select the functional class $\phi$ for the trend [19].

**Knowledge of Bound $\epsilon$ on the Distribution Shift:** Unlike the smoothly non-stationary setting, if the underlying shift can be discrete and arbitrary, prior data may not contain any useful information towards characterizing the shift. Therefore, avoiding domain knowledge may be inevitable when setting the value for $\epsilon$ unless some other stronger assumptions are made.

## C   Extended Discussion on Related Work

In the on-policy RL literature, parameters other than the mean have also been explored [44, 81, 21, 101, 27, 54, 4], and recent distributional RL methods extend this direction by estimating the entire distribution of returns [62, 63, 7, 22, 23, 24, 75]. Our work builds upon many of these ideas and extends them to the off-policy setting.

In the off-policy RL setup, there is a large body of literature that tackles the off-policy mean estimation problem [71, 83]. Some works also aim at providing high-confidence off-policy mean estimation using concentration inequalities [91, 53] or bootstrapping [92, 40, 51]. Several recent approaches build upon a dual perspective for dynamic programming [72, 100, 64] for both estimating and bounding the mean [57, 104, 45, 95, 25, 35]. However, these methods are restricted to domains with Markovian dynamics and full observability. Some works have also focused on estimating the mean return in the setting where states are partially observed [65, 85, 48] or when there is non-stationarity [16, 17, 50, 66]. Recent work by Chandak et al. [18] also looks at (high-confidence) off-policy variance estimation. Our work extends these research directions by tackling these settings simultaneously, while also providing a general procedure to estimate and obtain high-confidence bounds for *any* parameter of the distribution of returns. Particularly, UnO is a single, unified, and universal procedure that can be used to mitigate the complexity associated with estimating different parameters for different domain settings.

A popular RL method that has similar name to UnO is the *Universal value function approximator* (UVFA) by Schaul et al. [77]. However, UVFA is fundamentally different from UnO: UVFA estimates *expected* return $\mathbb{E}[G_\pi]$ from a state given any desired goal. By comparison, UnO estimates any parameter of the return $G_\pi$ for a single "goal". Recent work by Harb et al. [42] and Faccio et al. [33] propose using supervised learning to estimate parametric models that can map a *representation* of a policy $\pi$ to the corresponding distribution of $G_\pi$. By training over a given distribution of policies, new policies in the test set can be evaluated without using new data. By comparison, UnO does not requires any parametric assumptions or any train-test distribution. Further, UnO also provides high-confidence bounds for all the parameters of the return distribution.

# D Proofs for Theoretical Results

The main results in this paper are for the setting where both the evaluation and the behavior policies have the same observation set. In the following, we present generalized results where the available observations, $\widetilde{O}$, for the evaluation policy can be different from the behavior policy's observations, $O$. Further, for notational ease, in the main paper we had focused only on finite sets. In the following, we present a more general setting where states, actions, observations, and rewards are all continuous. Let $\Omega_2 : \mathcal{S} \times \mathcal{O} \to \Delta(\widetilde{\mathcal{O}})$ be the distribution over $\widetilde{\mathcal{O}}$, conditioned on state $s \in \mathcal{S}$ and observation $o \in \mathcal{O}$, which determines how the observations $\widetilde{O}$ are generated.

Let $\mathcal{D} = (H_i)_{i=1}^n$ be the available observed trajectories, where each $H$ contains $(\widetilde{O}_0, A_0, \beta(A_0|O_0), R_0, \widetilde{O}_1, ...)$. Note that when the random variables $\widetilde{O} = O = S$, we recover a standard fully observable MDP setting. By comparison, $H_\pi$ is the random variable corresponding to the complete trajectory $(S_0, O_0, \widetilde{O}_0, A_0, R_0, S_1, O_1, \widetilde{O}_1, ...)$ under any policy $\pi$. Of course, $H_\pi$ is unknown. To make the dependence between a trajectory $h \in \mathscr{H}_\pi$ and its associated return $G$ and importance ratios $\rho$ explicit, we use the shorthand $g(h)$ and $\rho(h)$ to denote the return and importance ratios for the full trajectory $h$, respectively. To tackle this generalized setting, we also generalize the support assumption introduced earlier,

**Assumption 1.** *The set $\mathcal{D}$ contains independent (not necessarily identically distributed) observed trajectories generated using $(\beta_i)_{i=1}^n$, such that for some (unknown) $\varepsilon > 0$, $(\beta(a|o) < \varepsilon) \implies (\pi(a|\tilde{o}) = 0)$, for all $s \in \mathcal{S}, o \in supp(\Omega(s)), \tilde{o} \in supp(\Omega_2(s,o)), a \in \mathcal{A}$, and $i \in \{1, \ldots, n\}$.*

**Theorem 1.** *Under Assumption 1, $\hat{F}_n$ is an unbiased and uniformly consistent estimator of $F_\pi$. That is,*

$$\forall \nu \in \mathbb{R}, \quad \mathbb{E}_{\mathcal{D}}\left[\hat{F}_n(\nu)\right] = F_\pi(\nu), \qquad \sup_{\nu \in \mathbb{R}} \left|\hat{F}_n(\nu) - F_\pi(\nu)\right| \xrightarrow{a.s.} 0.$$

*Proof.* This theorem has two results: unbiasedness and consistency of $\hat{F}_n$. Therefore, we break the proof into two parts.

**Part 1 (Unbiasedness).** We begin by expanding $F_\pi$ for any $\nu \in \mathbb{R}$ using the definition of the CDF.

$$F_\pi(\nu) = \Pr(G_\pi \le \nu) = \int_{-\infty}^\nu p(G_\pi = x)\mathrm{d}x = \int_{-\infty}^\nu \left(\int_{\mathscr{H}_\pi} p(H_\pi = h)\mathbb{1}_{\{g(h)=x\}}\mathrm{d}h\right)\mathrm{d}x, \quad (8)$$

where we used the fact that the probability density of the return $G_\pi$ being $x$ is the integral of the probability densities of the trajectories $h$ whose return equals $x$. Therefore, as the integrands in (8) are finite and non-negative measurable functions, using Tonelli's theorem for interchanging the integrals, (8) can be expressed as,

$$F_\pi(\nu) = \int_{\mathscr{H}_\pi} p(H_\pi = h)\left(\int_{-\infty}^\nu \mathbb{1}_{\{g(h)=x\}}\mathrm{d}x\right)\mathrm{d}h = \int_{\mathscr{H}_\pi} p(H_\pi = h)\left(\mathbb{1}_{\{g(h)\le\nu\}}\right)\mathrm{d}h, \quad (9)$$

where the last term follows because the output of $g(h)$ is a deterministic scalar given $h$ and thus the indicator function can be one for at most a single value less than $\nu$, and where the red color is used to highlight changes. Next, using Assumption 1 to change the support of the distribution in (9) and using importance weights we obtain,

$$F_\pi(\nu) = \int_{\mathscr{H}_\beta} p(H_\pi = h)\left(\mathbb{1}_{\{g(h)\le\nu\}}\right)\mathrm{d}h = \int_{\mathscr{H}_\beta} p(H_\beta = h)\frac{p(H_\pi = h)}{p(H_\beta = h)}\left(\mathbb{1}_{\{g(h)\le\nu\}}\right)\mathrm{d}h. \quad (10)$$

To simplify (10), we recursively use the fact that $p(X, Y) = p(X)p(Y|X)$ and note that under a given policy $\pi$ the probability density of a trajectory with partial observations and non-Markovian structure is

$$\begin{aligned}
p(H_\pi = h) = &p(s_0)p(o_0|s_0)p(\tilde{o}_0|o_0, s_0)p(a_0|s_0, o_0, \tilde{o}_0; \pi) \\
&\times \prod_{i=0}^{T-1} \Bigg( p(r_i|h_i)p(s_{i+1}|h_i)p(o_{i+1}|s_{i+1}, h_i)p(\tilde{o}_{i+1}|s_{i+1}, o_{i+1}, h_i) \\
&\qquad \times p(a_{i+1}|s_{i+1}, o_{i+1}, \tilde{o}_{i+1}, h_i; \pi) \Bigg) p(r_T|h_T),
\end{aligned} \quad (11)$$

where conditioning on $\pi$ emphasizes that each action is sampled using $\pi$, and $h_i$ represents the trajectory of all the states, partial observations, and actions up to time step $i$. Therefore, using (11), the ratio between $p(H_\pi = h)$ and $p(H_\beta = h)$ can be written as,

$$\frac{p(H_\pi = h)}{p(H_\beta = h)} = \frac{p(a_0|s_0, o_0, \tilde{o}_0; \pi)}{p(a_0|s_0, o_0, \tilde{o}_0; \beta)} \prod_{i=0}^{T-1} \frac{p(a_{i+1}|s_{i+1}, o_{i+1}, \tilde{o}_{i+1}, h_i; \pi)}{p(a_{i+1}|s_{i+1}, o_{i+1}, \tilde{o}_{i+1}, h_i; \beta)}$$

$$= \prod_{i=0}^{T} \frac{\pi(a_i|\tilde{o}_i)}{\beta(a_i|o_i)}$$

$$= \rho(h). \tag{12}$$

Combining (10) and (12),

$$F_\pi(\nu) = \int_{\mathscr{H}_\beta} p(H_\beta = h)\rho(h)\Big(\mathbb{1}_{\{g(h)\leq\nu\}}\Big)\mathrm{d}h. \tag{13}$$

Finally, it can be shown that our proposed estimator $\hat{F}_n$ is an unbiased estimator of $F_\pi$ by taking the expected value of $\hat{F}_n$,

$$\mathbb{E}_\mathcal{D}\Big[\hat{F}_n(\nu)\Big] = \mathbb{E}_\mathcal{D}\left[\frac{1}{n}\sum_{i=1}^{n} \rho_i\Big(\mathbb{1}_{\{G_i\leq\nu\}}\Big)\right]$$

$$= \frac{1}{n}\sum_{i=1}^{n} \mathbb{E}_\mathcal{D}\Big[\rho_i\Big(\mathbb{1}_{\{G_i\leq\nu\}}\Big)\Big]$$

$$= \frac{1}{n}\sum_{i=1}^{n} \int_{\mathscr{H}_{\beta_i}} p(H_{\beta_i} = h)\rho(h)\Big(\mathbb{1}_{\{g(h)\leq\nu\}}\Big)\mathrm{d}h$$

$$\overset{(a)}{=} \frac{1}{n}\sum_{i=1}^{n} F_\pi(\nu)$$

$$= F_\pi(\nu), \tag{14}$$

where (a) follows from (13), which holds for any behavior policy $\beta$ that satisfies Assumption 1.

**Note:** $H_\pi$ or $H_\beta$ were invoked only for the purposes of the proof. Notice that the proposed estimator, $\hat{F}_n(\nu) = \frac{1}{n}\sum_{i=1}^{n} \rho_i\big(\mathbb{1}_{\{G_i\leq\nu\}}\big)$, only depends on the quantities available in the observed trajectory $(H_i)_{i=1}^{n}$ from $\mathcal{D}$.

**Part 2 (Uniform Consistency).** For this part, we will first show pointwise consistency, i.e., for any $\nu$, $\hat{F}_n(\nu) \xrightarrow{\text{a.s.}} F_\pi(\nu)$, and then we will use this to establish *uniform* consistency, as required. To do so, let

$$X_i := \rho_i\Big(\mathbb{1}_{\{G_i\leq\nu\}}\Big).$$

From Assumption 1, we know that trajectories are independent and that $\beta(a|o) \geq \varepsilon$ when $\pi(a|\tilde{o}) > 0$. This implies that the denominator in the IS ratio is bounded below when $\pi(a|\tilde{o}) \neq 0$, and hence the $X_i$'s are bounded above and have a finite variance. Further, as established in (14), the expected value of $X_i$ for all $i$ equals $F_\pi(\nu)$. Therefore, using Kolmogorov's strong law of large numbers [78, Theorem 2.3.10 with Proposition 2.3.10],

$$\hat{F}_n(\nu) = \frac{1}{n}\sum_{i=1}^{n} X_i \xrightarrow{\text{a.s.}} \mathbb{E}_\mathcal{D}\left[\frac{1}{n}\sum_{i=1}^{n} X_i\right] = F_\pi(\nu). \tag{15}$$

In the following, to obtain uniform consistency, we follow the proof for the Glivenko-Cantelli theorem [38, 15, 79, 82] using the pointwise consistency of the off-policy CDF estimator $\hat{F}_n$ established in (15). The proof relies upon the construction of $K$ key points such that the difference in $F_\pi$ at successive key points is bounded by a small $\epsilon_1$. However, this would not be possible directly as there

can be discontinuties/jumps in $F_\pi$ that are greater than $\epsilon_1$. To tackle such discontinuties, we introduce some extra notation, Formally, let, $\forall \nu \in \mathbb{R}$,

$$F_\pi(\nu^-) := \Pr(G_\pi < \nu) = F_\pi(\nu) - \Pr(G_\pi = \nu), \qquad \hat{F}_n(\nu^-) := \frac{1}{n} \sum_{i=1}^{n} \rho_i \left( \mathbb{1}_{\{G_i < \nu\}} \right). \tag{16}$$

Then, using arguments analogous to the ones used for (15), it can be observed that

$$\hat{F}_n(\nu^-) \xrightarrow{\text{a.s}} F_\pi(\nu^-). \tag{17}$$

Let $\epsilon_1 > 0$, and let $K$ be any value more than $1/\epsilon_1$. Let $(\kappa_i)_{i=0}^{K}$ be $K$ key points,

$$G_{\min} = \kappa_0 < \kappa_1 \le \kappa_2 .... \le \kappa_{K-1} < \kappa_K = G_{\max},$$

which create $K$ intervals such that for all $i \in (1, ..., K-1)$,

$$F_\pi(\kappa_i^-) \le \frac{i}{K} \le F_\pi(\kappa_i).$$

Then by construction, if $\kappa_{i-1} < \kappa_i$,

$$F_\pi(\kappa_i^-) - F_\pi(\kappa_{i-1}) \le \frac{i}{K} - \frac{i-1}{K} = \frac{1}{K} < \epsilon_1. \tag{18}$$

Intuitively, as $F_\pi$ is monotonically non-decreasing, (18) restricts the intermediate values for any $F_\pi(\nu)$, to be within an $\epsilon_1$ distance of the CDF values at its nearby key points. Notice the role of $\kappa_i^-$ here: it would not have been possible to bound difference between $F_\pi(\kappa_i)$ and $F_\pi(\kappa_{i-1})$ by $\epsilon_1$ as there could have been 'jumps' of value greater than $\epsilon_1$ in $F_\pi$. However, $\kappa^-$ and $\kappa$ can be used to consider key points right before and after any jump in $F_\pi$, which ensures that we can always construct sequence of key points such that $F_\pi(\kappa_i^-) - F_\pi(\kappa_{i-1})$ is instead bounded by $\epsilon_1$.

For the CDF estimates at the key points, let,

$$\Delta_n := \max_{i \in (1...K-1)} \left\{ \left| \hat{F}_n(\kappa_i) - F_\pi(\kappa_i) \right|, \left| \hat{F}_n(\kappa_i^-) - F_\pi(\kappa_i^-) \right| \right\}. \tag{19}$$

From (15) and (17), as $\hat{F}_n(\nu)$ and $\hat{F}_n(\nu^-)$ are consistent estimators of $F_\pi(\nu)$ and $F_\pi(\nu^-)$, respectively, and since the maximum is over a finite set in (19), it follows that as $n \to \infty$,

$$\Delta_n \xrightarrow{\text{a.s}} 0. \tag{20}$$

For any $\nu$, let $\kappa_{i-1}$ and $\kappa_i$ be such that $\kappa_{i-1} \le \nu < \kappa_i$. Then,

$$\hat{F}_n(\nu) - F_\pi(\nu) \le \hat{F}_n(\kappa_i^-) - F_\pi(\kappa_{i-1})$$
$$\le \hat{F}_n(\kappa_i^-) - F_\pi(\kappa_i^-) + \epsilon_1, \tag{21}$$

where the last step follows using (18). Similarly,

$$\hat{F}_n(\nu) - F_\pi(\nu) \ge \hat{F}_n(\kappa_{i-1}) - F_\pi(\kappa_i^-)$$
$$\ge \hat{F}_n(\kappa_{i-1}) - F_\pi(\kappa_{i-1}) - \epsilon_1. \tag{22}$$

Then, using (21) and (22), $\forall \nu \in \mathbb{R}$,

$$\hat{F}_n(\kappa_{i-1}) - F_\pi(\kappa_{i-1}) - \epsilon_1 \le \hat{F}_n(\nu) - F_\pi(\nu) \le \hat{F}_n(\kappa_i^-) - F_\pi(\kappa_i^-) + \epsilon_1, \tag{23}$$

and thus using (19) and (23),

$$\left| \hat{F}_n(\nu) - F_\pi(\nu) \right| \le \Delta_n + \epsilon_1. \tag{24}$$

Using (20), we obtain the following property of the upper bound in (24):

$$\Delta_n + \epsilon_1 \xrightarrow{\text{a.s}} \epsilon_1. \tag{25}$$

Finally, since (24) holds for $\forall \nu \in \mathbb{R}$ and (25) is valid for any $\epsilon_1 > 0$, making $\epsilon_1 \to 0$ gives the desired result,

$$\sup_{\nu \in \mathbb{R}} \left| \hat{F}_n(\nu) - F_\pi(\nu) \right| \xrightarrow{\text{a.s.}} 0. \tag{26}$$

$\square$

**Variance-reduced estimation:** It is known that importance-sampling-based estimators are subject to high variance, which can often be limiting in practice [39]. A popular approach to mitigate variance is to use *weighted* importance sampling (WIS), which trades off variance for bias. Leveraging this approach, we propose the following variance-reduced estimator, $\bar{F}_n$, of $F_\pi$,

$$\forall \nu \in \mathbb{R}, \quad \bar{F}_n(\nu) := \frac{1}{\sum_{j=1}^n \rho_j} \left( \sum_{i=1}^n \rho_i \left( \mathbb{1}_{\{G_i \leq \nu\}} \right) \right). \tag{27}$$

In the following theorem, we show that $\bar{F}_n$ is a biased estimator of $F_\pi$, though it preserves consistency.

**Property 1.** *Under Assumption* 1, $\bar{F}_n$ *may be biased but is a uniformly consistent estimator of* $F_\pi$,

$$\forall \nu \in \mathbb{R}, \quad \mathbb{E}_{\mathcal{D}}\left[ \bar{F}_n(\nu) \right] \neq F_\pi, \qquad\qquad \sup_{\nu \in \mathbb{R}} \left| \bar{F}_n(\nu) - F_\pi(\nu) \right| \xrightarrow{a.s.} 0.$$

*Proof.* Similar to the proof for Theorem 1, we break this proof in two parts, one to establish bias and the other to establish consistency of $\hat{F}_n$.

**Part 1 (Biased):** We prove this using a counter-example. Let $n = 1$ and $\pi \neq \beta_1$, so

$$
\begin{aligned}
\forall \nu \in \mathbb{R}, \quad \mathbb{E}_{\mathcal{D}}\left[ \bar{F}_n(\nu) \right] &= \mathbb{E}_{\mathcal{D}}\left[ \frac{1}{\sum_{j=1}^1 \rho_j} \left( \sum_{i=1}^1 \rho_i \mathbb{1}_{\{G_i \leq \nu\}} \right) \right] \\
&= \mathbb{E}_{\mathcal{D}}\left[ \mathbb{1}_{\{G_1 \leq \nu\}} \right] \\
&\overset{(a)}{=} \int_{\mathscr{H}_{\beta_1}} p(H_{\beta_1} = h) \left( \mathbb{1}_{\{g(h) \leq \nu\}} \right) \mathrm{d}h \\
&= F_{\beta_1}(\nu) \\
&\neq F_\pi(\nu),
\end{aligned}
$$

where (a) follows analogously to (9).

**Part 2 (Uniform Consistency):** First, we will establish pointwise consistency, i.e., for any $\nu$, $\bar{F}_n(\nu) \xrightarrow{a.s.} F_\pi(\nu)$, and then we will use this to establish *uniform* consistency, as required.

$$
\begin{aligned}
\forall \nu \in \mathbb{R}, \quad \bar{F}_n(\nu) &= \frac{1}{\sum_{j=1}^1 \rho_j} \left( \sum_{i=1}^1 \rho_i \mathbb{1}_{\{G_i \leq \nu\}} \right) \\
&= \left( \frac{1}{n} \sum_{j=1}^n \rho_j \right)^{-1} \left( \frac{1}{n} \sum_{i=1}^n \rho_i \mathbb{1}_{\{G_i \leq \nu\}} \right).
\end{aligned}
$$

Let $X_n := \frac{1}{n} \sum_{j=1}^n \rho_j$ and $Y_n := \frac{1}{n} \sum_{i=1}^n \rho_i \mathbb{1}_{\{G_i \leq \nu\}}$. Now, as $\bar{F}_n(\nu)$ is a continuous function of both $X_n$ and $Y_n$, if both $\left( \lim_{n \to \infty} X_n \right)^{-1}$ and $\left( \lim_{n \to \infty} Y_n \right)$ exist then using the continuous mapping theorem [96, Theorem 2.3],

$$\forall \nu \in \mathbb{R}, \quad \lim_{n \to \infty} \bar{F}_n(\nu) = \left( \lim_{n \to \infty} X_n \right)^{-1} \left( \lim_{n \to \infty} Y_n \right). \tag{28}$$

Notice using Kolmogorov's strong law of large numbers [78, Theorem 2.3.10 with Proposition 2.3.10] that the term in the first parentheses will almost surely converge to the expected value of importance ratios, which equals one [71]. Similarly, we know from (15) that the term in the second parentheses will converge to $F_\pi(\nu)$ almost surely. Therefore, both parenthetical terms of (28) exist, and thus

$$\forall \nu \in \mathbb{R}, \quad \bar{F}_n(\nu) \xrightarrow{a.s.} (1)^{-1}(F_\pi(\nu)) = F_\pi(\nu). \tag{29}$$

Now, similar to the proof for Theorem 1, combining (29) with arguments from (16) to (26), it can be observed that

$$\sup_{\nu \in \mathbb{R}} \left| \bar{F}_n(\nu) - F_\pi(\nu) \right| \xrightarrow{a.s.} 0.$$

$\square$

**Theorem 2.** *Under Assumption 1, for any $\delta \in (0, 1]$, if $\sum_{i=1}^{K} \delta_i \leq \delta$, then the confidence band defined by $F_-$ and $F_+$ provides guaranteed coverage for $F_\pi$. That is,*

$$\Pr\left(\forall \nu, \ F_-(\nu) \leq F_\pi(\nu) \leq F_+(\nu)\right) \geq 1 - \delta.$$

*Proof.* Let $A_i$ be the event that for the key point $\kappa_i$, $\mathtt{CI}_-(\kappa_i, \delta_i) \leq F_\pi(\kappa_i) \leq \mathtt{CI}_+(\kappa_i, \delta_i)$, for all $i \in (1, ..., K)$. Let superscript $c$ denote a complementary event; then by the union bound, the total probability of the bounds holding at each key point simultaneously is

$$\Pr\left(\cap_{i=1}^{K} A_i\right) = 1 - \Pr\left((\cap_{i=1}^{K} A_i)^c\right) = 1 - \Pr\left(\cup_{i=1}^{K} A_i^c\right) \geq 1 - \sum_{i=1}^{K} \Pr\left(A_i^c\right) \overset{(a)}{\geq} 1 - \delta, \text{(30)}$$

where $(a)$ holds because the conditions of the theorem assert that the sum of probabilities of the bounds failing at each key point is at most $\delta$. Therefore, using (30),

$$\Pr\left(\forall i \in (1, ..., K), \ \mathtt{CI}_-(\kappa_i, \delta_i) \leq F_\pi(\kappa_i) \leq \mathtt{CI}_+(\kappa_i, \delta_i)\right) \geq 1 - \delta. \quad (31)$$

Since by construction, at the key points $(\kappa_i)_{i=1}^{K}$, $F_-(\kappa_i) = \mathtt{CI}_-(\kappa_i, \delta_i)$ and $F_+(\kappa_i) = \mathtt{CI}_+(\kappa_i, \delta_i)$, it follows from (31) that

$$\Pr\left(\forall i \in (1, ..., K), \ F_-(\kappa_i) \leq F_\pi(\kappa_i) \leq F_+(\kappa_i)\right) \geq 1 - \delta. \quad (32)$$

Using the monotonically non-decreasing property of a CDF, at any point $\nu \in \mathbb{R}$ such that $\kappa_i \leq \nu \leq \kappa_{i+1}$, we know that $F_\pi(\kappa_i) \leq F_\pi(\nu) \leq F_\pi(\kappa_{i+1})$. Therefore, when the bounds at the key points hold, $F_\pi$ at the key points can also be upper and lower bounded: $F_-(\kappa_i) \leq F_\pi(\nu) \leq F_+(\kappa_{i+1})$. Therefore, by (32) and the construct in (6), it immediately follows that

$$\Pr\left(\forall \nu, \ F_-(\nu) \leq F_\pi(\nu) \leq F_+(\nu)\right) \geq 1 - \delta.$$

$\square$

**Theorem 3.** *Under Assumption 1, for any $1 - \delta$ confidence band $\mathcal{F}$, the confidence interval defined by $\psi_-$ and $\psi_+$ provides guaranteed coverage for $\psi(F_\pi)$. That is,*

$$\Pr\left(\psi_- \leq \psi(F_\pi) \leq \psi_+\right) \geq 1 - \delta.$$

*Proof.* Recall that the confidence band $\mathcal{F}$ is a random variable dependent on the data $\mathcal{D}$. Let $\mathbb{E}_\mathcal{F}[\cdot]$ represent expectation with respect to $\mathcal{F}$, then repeatedly using the law of total probability,

$$\begin{aligned}
\Pr\left(\psi_- \leq \psi(F_\pi) \leq \psi_+\right) &= \mathbb{E}_\mathcal{F}\left[\Pr\left(\psi_- \leq \psi(F_\pi) \leq \psi_+ \middle| \mathcal{F}\right)\right] \\
&= \mathbb{E}_\mathcal{F}\Big[\Pr\left(\psi_- \leq \psi(F_\pi) \leq \psi_+ \middle| F_\pi \in \mathcal{F}, \mathcal{F}\right) \Pr\left(F_\pi \in \mathcal{F} \middle| \mathcal{F}\right) \\
&\qquad + \Pr\left(\psi_- \leq \psi(F_\pi) \leq \psi_+ \middle| F_\pi \notin \mathcal{F}, \mathcal{F}\right) \Pr\left(F_\pi \notin \mathcal{F} \middle| \mathcal{F}\right)\Big] \\
&\geq \mathbb{E}_\mathcal{F}\left[\Pr\left(\psi_- \leq \psi(F_\pi) \leq \psi_+ \middle| F_\pi \in \mathcal{F}, \mathcal{F}\right) \Pr\left(F_\pi \in \mathcal{F} \middle| \mathcal{F}\right)\right] \\
&\overset{(a)}{=} \mathbb{E}_\mathcal{F}\left[\Pr\left(F_\pi \in \mathcal{F} \middle| \mathcal{F}\right)\right] \\
&= \Pr\left(F_\pi \in \mathcal{F}\right) \\
&\overset{(b)}{\geq} 1 - \delta,
\end{aligned}$$

where $(a)$ follows from that fact that $F_\pi \in \mathcal{F}$ implies $\psi_- \leq \psi(F_\pi) \leq \psi_+$. Step $(b)$ follows from Theorem 2. $\square$

*Proof (Alternate).* This proof is shorter but requires a theoretical construct of a *set of sets of functions*. That is, let $\mathbb{F}$ be any set of cumulative distribution functions and $\mathscr{F}$ be a set of such sets, such that

$$\mathscr{F} := \left\{\mathbb{F} \ \middle| \ F_\pi \in \mathbb{F}\right\}.$$

In other words, $\mathbb{F}$ is the set of CDFs which contains the true CDF $F_\pi$, and $\mathscr{F}$ is the set of *all* such sets $\mathbb{F}$. From Theorem 2, we know that the confidence band $\mathcal{F}$ contains $F_\pi$ with probability at least $1 - \delta$. Therefore, it also holds that

$$\Pr(\mathcal{F} \in \mathscr{F}) \geq 1 - \delta.$$

However, the event $(\mathcal{F} \in \mathscr{F})$ implies that $\psi_- \leq \psi(F_\pi) \leq \psi_+$ as $F_\pi$ is contained in this specific $\mathcal{F}$ used to construct $\psi_-$ and $\psi_+$. Therefore, it also holds that

$$\Pr(\psi_- \leq \psi(F_\pi) \leq \psi_+) \geq 1 - \delta.$$

$\square$

**Theorem 4.** *Under Assumptions 1 and 2, for any $\delta \in (0, 1]$, the confidence band defined by $F_-^{(2)}$ and $F_+^{(2)}$ provides guaranteed coverage for $F_\pi^{(2)}$. That is,*

$$\Pr\left(\forall \nu, \ F_-^{(2)}(\nu) \leq F_\pi^{(2)}(\nu) \leq F_+^{(2)}(\nu)\right) \geq 1 - \delta.$$

*Proof.* From Assumption 2, $\sup\limits_{\nu \in \mathbb{R}} \left| F_\pi^{(1)}(\nu) - F_\pi^{(2)}(\nu) \right| \leq \epsilon$. Or equivalently,

$$\forall \nu \in \mathbb{R}, \quad F_\pi^{(1)}(\nu) - \epsilon \leq F_\pi^{(2)}(\nu) \leq F_\pi^{(1)}(\nu) + \epsilon. \tag{33}$$

Using Theorem 2 for the bound obtained on $F_\pi^{(1)}$ for the first domain,

$$\Pr\left(\forall \nu, \ F_-^{(1)}(\nu) \leq F_\pi^{(1)}(\nu) \leq F_+^{(1)}(\nu)\right) \geq 1 - \delta. \tag{34}$$

Therefore, combining (33) and (34),

$$\Pr\left(\forall \nu, \ F_-^{(1)}(\nu) - \epsilon \leq F_\pi^{(2)}(\nu) \leq F_+^{(1)}(\nu) + \epsilon\right) \geq 1 - \delta. \tag{35}$$

Then by the construct in (7), it follows from (35) that

$$\Pr\left(\forall \nu, \ F_-^{(2)}(\nu) \leq F_\pi^{(2)}(\nu) \leq F_+^{(2)}(\nu)\right) \geq 1 - \delta.$$

$\square$

# E   Extended Discussion for UnO

## E.1   Nuances for CDF Inverse and CVaR

For brevity, some nuances for $\hat{F}_n^{-1}(\alpha)$ and $\mathrm{CVaR}_\pi^\alpha(\hat{F}_n)$ were excluded from the main paper. We discuss them in this section.

As discussed earlier in Remark 1, it is possible that $\hat{F}_n(\nu) > 1$ for some $\nu \in \mathbb{R}$ due to the use of importance weighting. Similarly, it is also possible that $\hat{F}_n(\nu) < 1$ for all $\nu \in \mathbb{R}$. Specifically, if $\hat{F}_n(\nu) < \alpha$ for all $\nu$, then it raises the question: how can one obtain an estimate of $F_\pi^{-1}(\alpha)$? To resolve this issue, we use the following estimator of $F_\pi^{-1}(\alpha)$ for UnO:

$$\hat{F}_n^{-1}(\alpha) := \begin{cases} \min\left\{g \in (G_{(i)})_{i=1}^n \Big| \hat{F}_n(g) \geq \alpha\right\}, & \text{if} \quad \exists g \text{ s.t. } \hat{F}_n(g) \geq \alpha, \\ \max(G_{(i)})_{i=1}^n & \text{otherwise.} \end{cases}$$

However, it is known from Theorem 1 that $\hat{F}_n$ is a uniformly consistent estimator of $F_\pi$. Therefore, the edge case that $\hat{F}_n(\nu) < \alpha$ for all $\nu$ cannot occur in the limit as $n \to \infty$. Resolving this is required mostly when the sample size is small.

Regarding CVaR, it is known [1] that when the distribution of a random variable (which is $G_\pi$ for UnO) is continuous, then CVaR can be expressed as,

$$\mathrm{CVaR}_\pi^\alpha(F_\pi) = \mathbb{E}\left[G_\pi | G_\pi \leq F_\pi^{-1}(\alpha)\right], \tag{36}$$

and thus an off-policy sample estimator for (36) can be constructed as,

$$\text{CVaR}_\pi^\alpha(\hat{F}_n) := \frac{1}{\alpha} \sum_{i=1}^n \mathrm{d}\hat{F}_n(G_{(i)}) G_{(i)} \mathbb{1}_{\left\{G_{(i)} \le Q_\pi^\alpha(\hat{F}_n)\right\}}.$$

However, for distributions that are not continuous, a more generic definition for CVaR is [13],

$$\text{CVaR}_\pi^\alpha(F_\pi) = \inf_g \left\{ g - \frac{1}{\alpha} \mathbb{E}\Big[ \max\big(0, g - G_\pi\big) \Big] \right\}. \tag{37}$$

We extend the sample estimator by Brown [13] for (37) and use the following off-policy estimator for UnO:

$$\text{CVaR}_\pi^\alpha(\hat{F}_n) := \hat{F}_n^{-1}(\alpha) - \frac{1}{\alpha} \sum_{i=1}^n \mathrm{d}\hat{F}_n(G_{(i)}) \left( \max\big(0, \hat{F}_n^{-1}(\alpha) - G_{(i)}\big) \right)$$

## E.2 Optimizing Confidence Bands for Tighter Bounds:

Constructing $\mathcal{F}$ requires selecting $K$ key points for which CIs are computed. If too many key points are selected, then each $\delta_i$ has to be a very small positive value so that $\sum_{i=1}^K \delta_i \le \delta$, as required by Theorem 2. This will make the confidence intervals wide at each key point. In contrast, if too few key points are selected, then the confidence intervals at the $\kappa_i$'s will be relatively tighter, but this will not tighten the intervals *between* the $\kappa_i$'s due to the way $F_-$ and $F_+$ are constructed in (5). Further, the overall tightness of $\mathcal{F}$ is also affected by the location of each $\kappa_i$ and its respective failure rate $\delta_i$. Therefore, to get a tight $\mathcal{F}$, we propose searching for a $\theta := \big(K, (\kappa_i)_{i=1}^K, (\delta_i)_{i=1}^K\big)$ that minimizes the area enclosed in $\mathcal{F}$. That is, let $\Delta_{i+1} := \kappa_{i+1} - \kappa_i$, then the area enclosed in $\mathcal{F}$ is

$$\mathscr{A}(\theta) := \sum_{i=0}^K \big(\texttt{CI}_+(\kappa_{i+1}, \delta_{i+1}) - \texttt{CI}_-(\kappa_i, \delta_i)\big) \Delta_{i+1}.$$

To avoid multiple comparisons [8], we first partition $\mathcal{D}$ into $\mathcal{D}_\text{train}$ and $\mathcal{D}_\text{eval}$. Subsequently, $\mathcal{D}_\text{train}$ is used to search for $\theta^*$ as follows, and then $\theta^*$ is used with $\mathcal{D}_\text{eval}$ to obtain $\mathcal{F}$.

$$\theta^* := \underset{\theta}{\arg\min}\ \mathscr{A}(\theta) \tag{38}$$

$$\text{s.t.} \qquad G_\text{min} < \kappa_i < G_\text{max}, \quad \sum_{i=1}^K \delta_i \le \delta, \quad \delta_i \ge 0, \qquad \forall i \in (1, ..., K).$$

**Remark 5.** *A global optimum of (38) is not required—any feasible $\theta$ can be used with $\mathcal{D}_{eval}$ to obtain a confidence band $\mathcal{F}$. Optimization only helps by making the band tighter.*

For our experimental results, when searching $\theta^*$ for (38), we keep the number of key points, $K$, fixed to $\log(n)$, where $n$ is the number of observed trajectory samples in $\mathcal{D}$. To search for the locations $(\kappa_i)_{i=1}^K$ and the failure rates $(\delta_i)_{i=1}^K$ at each key point, we use the BlackBoxOptim library[2] available in Julia [11]. To perform this optimization, we construct $\mathcal{D}_\text{train}$ using $5\%$ of data from $\mathcal{D}$, and construct $\mathcal{D}_\text{eval}$ using the rest of the data. Following the idea by Thomas et al. [91], when searching for $\theta^*$ using $\mathcal{D}_\text{train}$, bounds for the key points $(\kappa_i)_{i=1}^K$ are obtained as if the number of samples are equal to the number of samples available in $\mathcal{D}_\text{eval}$ (see Equation 7 in the work by Thomas et al. [91] for more discussion on this). Instead of using a single split, one could potentially also leverage results by Romano and DiCiccio [73] to use multiple splits; we leave this for future work.

## E.3 Bound Specialization

In (38), $\theta$ was searched to minimize the area $\mathscr{A}(\theta)$ enclosed within $\mathcal{F}(\theta)$, where $\mathcal{F}(\theta)$ represents the CDF band obtained using the parameter $\theta$. This was done without any consideration of the downstream parameter $\psi$ for which the bounds would be constructed using $\mathcal{F}(\theta)$. Therefore, the

---

[2]https://github.com/robertfeldt/BlackBoxOptim.jl

band $\mathcal{F}(\theta)$ is tight overall, but need not be the best possible if only a specific parameter $\psi$'s bounds are required using $\mathcal{F}(\theta)$.

For example, consider obtaining bounds for CVaR$_\pi^\alpha$. As can be seen from the geometric insight in Figure 3, bounds for CVaR are mostly dependent on the tightness of $\mathcal{F}(\theta)$ near the lower tail. Therefore, if one can obtain $\mathcal{F}(\theta)$ that is tighter near the lower tail, albeit looser near the upper tail, that would provide a better bound for CVaR as opposed to a band $\mathcal{F}(\theta)$ that has uniform tightness throughout.

To get a tight $\mathcal{F}(\theta)$ in such cases where there is a single downstream parameter of interest, we propose searching for a $\theta := \left(K, (\kappa_i)_{i=1}^K, (\delta_i)_{i=1}^K\right)$ that directly optimizes for the final parameter of interest instead of the area enclosed in $\mathcal{F}(\theta)$. For example, if only the lower bound for $\psi(F_\pi)$ is required, then let

$$\psi_-(\theta) := \inf_{F \in \mathcal{F}(\theta)} \psi(F).$$

Next, the optimization using $\mathcal{D}_{\text{train}}$ can then be modeled as the following,

$$\theta^* := \arg\max_\theta \psi_-(\theta)$$

$$\text{s.t.} \quad G_{\min} < \kappa_i < G_{\max}, \qquad\qquad \forall i \in (1, ..., K),$$

$$\sum_{i=1}^K \delta_i \leq \delta, \quad \delta_i \geq 0, \qquad\qquad \forall i \in (1, ..., K),$$

This would result in $\theta^*$ that when used with $\mathcal{D}_{\text{eval}}$ can be expected to provide the CDF band which will yield the highest lower bound for $\psi(F_\pi)$.

### E.4 Approximate Bounds for Any Parameter using Bootstrap

In Algorithm 1, we provide the pseudo code for obtaining bootstrap-based bounds for any parameter $\psi(F_\pi)$. In Line 1, $B$ datasets $(\mathcal{D}_i^*)_{i=1}^B$ are generated from $\mathcal{D}$ using resampling, and for each of these resampled data sets, $B$ (weighted IS-based) CDF estimates $(\bar{F}_{n,i}^*)_{i=1}^B$ are obtained. In Line 3, sample estimates $(\psi(\bar{F}_{n,i}^*))_{i=1}^B$ for the desired parameter $\psi(F_\pi)$ are constructed using the $B$ estimated CDFs. In Line 4, these sample estimates for $\psi(F_\pi)$ can be subsequently passed to the bias-corrected and accelerated (BCa [32]) bootstrap procedure to obtain approximate lower and upper bounds $(\psi_-, \psi_+)$.

---

**Algorithm 1:** Bootstrap Bounds for $\psi(F_\pi)$

---

1 **Input:** Dataset $\mathcal{D}$, Confidence level $1 - \delta$
2 Bootstrap $B$ datasets $(\mathcal{D}_i^*)_{i=1}^B$ and create $(\bar{F}_{n,i}^*)_{i=1}^B$
3 Bootstrap estimates $(\psi(\bar{F}_{n,i}^*))_{i=1}^B$ using $(\bar{F}_{n,i}^*)_{i=1}^B$
4 Compute $(\psi_-, \psi_+)$ using BCa$((\psi(\bar{F}_{n,i}^*))_{i=1}^B, \delta)$
5 **Return** $(\psi_-, \psi_+)$

---

### E.5 Extended Discussion of High-Confidence Bounds for Any Parameter

Section 4 of the main paper discussed how high-confidence bounds $\psi_-$ and $\psi_+$ can be obtained for any parameter $\psi(F_\pi)$ using the confidence band $\mathcal{F}$. Specifically, in Figure 3, geometric insights for obtaining the analytical form of the bounds for the mean, quantile, and CVaR were discussed. Extending that discussion, Figure 6 provides geometric insights for bounding other parameters, namely variance, inter-quantile ranges, and entropy, in the off-policy setting.

An advantage of having the CDF band $\mathcal{F}$ is that it can permit bounding other novel parameters that might be of interest. While analytical bounds using geometric insights, as discussed for a number of popular parameters, should also be the first attempt for the desired novel parameter, it may be the case that such geometric insight cannot be obtained. In such cases, a CDF $F$ can be directly parameterized using a spline curve, or a piecewise non-decreasing function that is constrained to be within $\mathcal{F}$. Depending on how rich this parameterization is, it may be feasible to use a black-box optimization routine and obtain a globally optimal $F$ that minimizes (maximizes) the desired parameter $\psi(F)$. If not feasible, an approximate bound can be achieved by using the best found local optima.

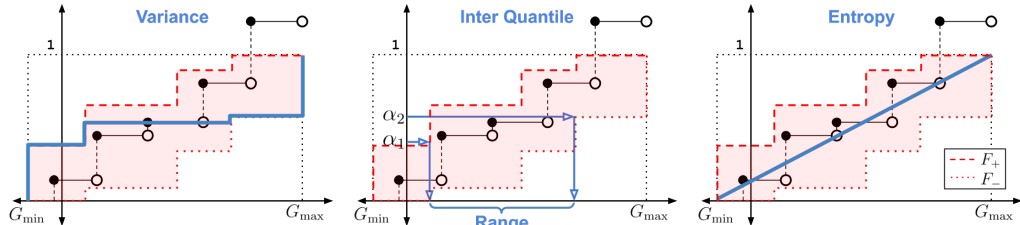

Figure 6: Similar to Figure 3, given a confidence band $\mathcal{F}$, lower and upper bounds for several other parameters can also be obtained using simple geometric insights. **(Left)** An upper bound for the variance can be obtained by observing that variance is maximized when the probability of events on either extreme are maximized. Therefore, the CDF $F \in \mathcal{F}$ for such a distribution will initially follow (from left to right) $F_+$ and then make a horizontal jump (at a specific jump point) to $F_-$, which it then follows until 1. The variance of the distribution with this CDF, $F$, will give the desired upper bound. Analogously, the CDF that initially follows $F_-$ and then jumps vertically (at a specific jump point) to $F_+$, assigns highest probability to events near the mean and thus results in the lowest variance [74]. **(Middle)** An upper bound for the inter-quantile range can be obtained by maximizing the value of upper $\alpha_2$-quantile and subtracting the minimum value for the lower $\alpha_1$-quantile. This can be obtained by $F_-^{-1}(\alpha_2) - F_+^{-1}(\alpha_1)$. Analogously, a lower bound can be obtained using $\max(0, F_+^{-1}(\alpha_2) - F_-^{-1}(\alpha_1))$. **(Right)** An upper bound on the entropy can be obtained by what Learned-Miller and DeStefano [55] call a "string-tightening" algorithm. That is, if the ends of a tight string are held at the bottom-left and the upper-right corner of $\mathcal{F}$, and the entire string is constrained to be within $\mathcal{F}$, then the path of the string corresponds to the $F \in \mathcal{F}$ that has highest entropy. In our figure, such an $F$ corresponds to the CDF of the uniform distribution, which is known to have maximum entropy. Unless some stronger assumptions are made, the lower bound on differential entropy is typically $-\infty$ if there is any possibility of a point mass.

## E.6 Tackling Smooth Non-stationarity using Wild Bootstrap

From Theorem 1, it is known that the proposed estimator $\hat{F}_n(\kappa)$ provides unbiased estimates for $F_\pi(\kappa)$, even with a single observed trajectory. In the non-stationary setting, let the true underlying CDF of returns for $\pi$ in the episode $i$ be $F_\pi^{(i)}(\kappa)$, and the estimate of $F_\pi^{(i)}(\kappa)$ using the trajectory observed during the episode $i$ be

$$\hat{F}_n^{(i)}(\kappa) := \rho_i \mathbb{1}_{\{G_i \leq \kappa\}} \qquad \forall i \in \{1, 2, ..., L\}.$$

Next, the trend of the sequence $\left(\hat{F}_n^{(i)}(\kappa)\right)_{i=1}^L$ can be analyzed to forecast $\hat{F}_n^{(L+\ell)}(\kappa)$ for the future episode $L + \ell$ when the policy $\pi$ will be executed. Particularly, under Assumption 3, $\exists w_\kappa$, such that, $\forall i \in (1, ..., L+\ell)$, $F_\pi^{(i)}(\kappa) = \phi(i)^\top w_\kappa$. Therefore, using the unbiased estimates $\left(\hat{F}_n^{(i)}(\kappa)\right)_{i=1}^L$ of $\left(F_\pi^{(i)}(\kappa)\right)_{i=1}^L$, we propose searching for $w_\kappa$ using least-squares regression. Let $X := [1, 2, ...., L]$ be the episode numbers in the past, then the predicates $\Phi_\kappa$, the targets $Y_\kappa$, and the corresponding least-squares solution $w_\kappa$ can be obtained as,

$$\Phi_\kappa := [\phi(X_1), \phi(X_2), ..., \phi(X_L)] \qquad \in \mathbb{R}^{L \times d},$$
$$Y_\kappa := [\hat{F}_n^{(1)}(\kappa), \hat{F}_n^{(2)}(\kappa), ..., \hat{F}_n^{(L)}(\kappa)] \qquad \in \mathbb{R}^{L \times 1},$$
$$w_\kappa := \left(\Phi_\kappa^\top \Phi_\kappa\right)^{-1} \Phi_\kappa^\top Y_\kappa \qquad \in \mathbb{R}^{d \times 1}.$$

Using $w_\kappa$, an unbiased estimate of $F_\pi^{(L+\ell)}(\kappa)$ can be obtained as,

$$\hat{F}_n^{(L+\ell)}(\kappa) := \phi(L + \ell)^\top w_\kappa. \tag{39}$$

The point forecast $\hat{F}_n^{(L+\ell)}(\kappa)$ from (39) can then be combined with Algorithms 1 and 2 presented by Chandak et al. [16] to obtain wild-bootstrap-based confidence intervals for $F_\pi^{(L+\ell)}(\kappa)$. Once the confidence intervals are obtained at different key points, (5) can be used to construct an entire confidence band for $F_\pi^{(L+\ell)}$.

# F Empirical Details

## F.1 Domain Details

In this section, we discuss domain details and how $\pi$ and $\beta$ were selected for these domains. The code for the domains, baselines [91, 18], and the proposed UnO estimator can be found at https://github.com/yashchandak/UnO.

**Recommender System:** Systems for online recommendation of tutorials, movies, advertisements, etc., are ubiquitous [86, 88]. In these settings, it may be beneficial to fully characterize a customer's experience once the new system/policy is deployed. To abstract such settings, we created a simulated domain where the user's interest for a finite set of items is represented using the corresponding item's reward.

Using an actor-critic algorithm [83], we find a near-optimal policy $\pi$, which we use as the evaluation policy. Let $\pi^{\text{rand}}$ be a random policy with uniform distribution over the actions (items). Then for an $\alpha = 0.5$, we define the behavior policy $\beta(a|s) := \alpha\pi(a|s) + (1-\alpha)\pi^{\text{rand}}(a|s)$ for all states and actions.

**Gridworld:** We also consider a standard continuous-state Gridworld with partial observability (which also makes the domain non-Markovian in the observations), stochastic transitions, and eight discrete actions corresponding to up, down, left, right, and the four diagonal movements. The off-policy data was collected using two different behavior policies, $\beta_1$ and $\beta_2$, and the evaluation policies for this domain were obtained similarly as for the recommender system domain discussed above. Particularly, using $\alpha = 0.5$, we define $\beta_1(a|o) := \alpha\pi(a|0) + (1-\alpha)\pi^{\text{rand}}(a|o)$ for all states and actions. Similarly, $\beta_2$ was defined using $\alpha = 0.75$.

**Diabetes Treatment:** This domain is modeled using an open source implementation [103] of the U.S. Food and Drug Administration (FDA) approved Type-1 Diabetes Mellitus Simulator (T1DMS) [59] for the treatment of type-1 diabetes. An episode corresponds to a day, and each step of an episode corresponds to a minute in an *in silico* patient's body and is governed by a continuous time nonlinear ordinary differential equation (ODE) [59]. In such potentially critical medical applications, it is important to go beyond just the expected performance and to characterize the risk associated with it, *before deployment*.

To control the insulin injection, which is required for regulating the blood glucose level, we use a policy that controls the parameters of a *basal-bolus controller*. This controller is based on the amount of insulin that a person with diabetes is instructed to inject prior to eating a meal [6]:

$$\text{injection} = \frac{\text{current blood glucose} - \text{target blood glucose}}{CF} + \frac{\text{meal size}}{CR},$$

where "current blood glucose" is the estimate of the person's current blood glucose level, "target blood glucose" is the desired blood glucose, "meal size" is the estimate of the size of the meal the patient is about to eat, and $CR \in [CR_{\min}, CR_{\max}]$ and $CF \in [CF_{\min}, CF_{\max}]$ are two parameters of the controller that must be tuned based on the body parameters to make the treatment effective. We designed an RL policy that acts on the discretized space of the parameters, $CR$ and $CF$, for the above basal-bolus controller. Behavior and evaluation policies were selected similarly as discussed for the recommender system domain.

## F.2 Extended Discussion on Results for Stationary Settings

The main results for the stationary setting are provided in Figure 4 of the main body. In this section, we provide some additional discussion on the observed trends for the bounds.

Notice in Figure 4 that UnO-CI bounds for the variance can require up to an order of magnitude less data compared to the existing bound for the variance [18]. This can be attributed to the fact that Chandak et al. [18] construct the bounds using $\mathbb{E}[\rho G^2] - \mathbb{E}[\rho G]^2$, where it can be observed that the second term depends quadratically on $\rho$. This makes the variance of that term effectively "doubly exponential" in the horizon length. This does not happen in the CDF-based approach as the bounds at any key point $\kappa$ depend on $\mathbb{E}[\rho\mathbb{1}_{G<\kappa}]$), which does not have any higher powers of $\rho$.

Another thing worth noting in Figure 4 is that not only the bounds for different parameters, but even the upper and lower bounds for the same parameter converge at different rates (especially for smaller values of $n$). Therefore, there are two particular trends to observe: (a) how close the bounds are to the true value at the beginning, and (b) how quickly they improve. Both of these depend on the direction for which clipping plays a major role and also how the bounds depend on the tails. For example, for the mean, as the distributions are right skewed (because evaluating policy $\pi$ is a near-optimal policy), the bounds on the CDF are clipped more from the lower end (so that $F(\nu) >= 0$ always). Therefore, since the upper bound on the mean depends on the lower CDF bound (see Figure 3), it starts close to the estimate itself but the progress actually seems slow because shrinking CDFs bounds at any specific $F(\nu)$ from the lower end does not impact the bound until the point where clipping is not required anymore.

For variance, the upper bound depends on both the upper bound on the lower tail and the lower bound on the upper tail (see Figure 6), and these two benefit from clipping the least and also converge the slowest. In contrast, the lower bound for variance depends on the upper bound on the upper tail and the lower bound on the lower tail, which are clipped immediately to be below 1 and above 0, respectively. Appendix B.1 (knowledge of $G_{\min}, G_{\max}$) and Fig 6 provide more intuition on this.