# OpenReview forum: "Universal Off-Policy Evaluation"
_NeurIPS.cc/2021/Conference — NeurIPS 2021 Poster_

### Official Review · Reviewer_j4b9 · 2021-07-15

**Rating:** 5
**Confidence:** 4

**Summary:**

Follow-up:
I check the paper again and now I am persuaded that the proof can be extended to the reward with continuous distribution. There seems no fundamental technical barrier to me. So I deem this issue as resolved and add 1 point.
However, my concern regarding the significance of the paper remains unchanged. The estimator proposed in the paper is based on the importance ratio and estimated CDF which is not novel. I also do NOT think the universality of such an estimator is a big contribution. (It is quite obvious that universality can follow from such estimators.)
Many significant previous works on OPE have been focused on proposing estimators with important properties, such as doubly robustness (e.g. https://arxiv.org/abs/1511.03722), free of curse of horizon (e.g. https://arxiv.org/pdf/1810.12429.pdf), etc. So there are many other properties that is more meaningful and applicable than a mere universality. An estimator that is only universal but does not come with any other desired and important properties is not an interesting estimator and will be of limited use in practice.

Therefore, I think the universality claimed by this paper is not interesting and significant enough. I feel this paper's significance is a little overstated.

============================
This paper studied the OPE problem for sequential decision-making and proposed an estimator that estimates and bounds the entire distribution of returns. The method is model-free and can obtain high confidence bounds for the parameters of the distribution. The method works under different settings of MDP: fully observable MDP vs POMDP; Markovian vs non-Markovian.


**Limitations And Societal Impact:**

Well addressed.

**Main Review:**

Originality: This paper is original. It is the first to study the estimation of a various of parameters of the reward distribution as far as I can tell.

Quality: The quality is OK. Theoretical results have clear proof and empirical results are well-explained.

Clarity: The writing is very clear and easy for the readers to understand.

Significance: The significance of this paper is affected by several factors. First, this paper assumes the return distribution is discrete. This assumption is too strong and restrictive, and the proof of the main results of the paper relies on this assumption. This assumption greatly simplifies the analysis and limits the usefulness of the results. Does the uniformity still hold under the general distribution assumption? As a comparison, in most of the related work, the only assumption on the reward distribution is usually a sub-Gaussian noise. Second, the proposed estimator hat{F} is naive. I suspect that it will suffer from the curse of the horizon. Still, this is avoided under the assumption of a discrete return distribution.
Third, the main Theorem 2 and 3 rely on knowing the confidence interval defined below Line 161. However, in the general case finding these CIs is itself the major challenge. Therefore, I think all the main results rely on strong assumptions/oracles, which greatly hurts the significance of the results. It is only universal under very simple settings and has very limited usage.



**Time Spent Reviewing:**

5

---

> ### Author Response · Authors · 2021-08-10
> **Response to reviewer j4b9**
>
> Thank you for your insightful comments. It has made us realize that we could have better phrased some of the paragraphs to avoid any confusion for the readers.
>
> > “this paper assumes the return distribution is discrete”
>
> We should have, and will move the footnote from page 3 to the notation section to clarify that a discrete reward distribution is not necessary, and Theorems 1, 2, 3, and 4 hold for both discrete and continuous distributions. The only differences in the theorems and proofs when considering the discrete and continuous settings is the use of integrals and probability density functions versus the use of summations and probability mass functions. Rather than write very similar versions of each theorem and proof, we opted to select one setting for our notation, with a footnote indicating that the conversion to the other setting is straightforward.
>
> We see that this can cause confusion and should be explained more clearly. We will explain this more clearly in the main paper by expanding the footnote and moving it into the main text. For completeness, we will also include the proofs for the continuous setting in the appendix.
>
> Further, it is worth highlighting that the proposed estimator is model-free and thus in practice can work directly with continuous observations, actions, and reward distributions. For example, In Fig 4 the diabetes treatment domain has both continuous states and a continuous distribution of returns, and the POMDP domain has continuous states.
>
> > “suffer from the curse of the horizon”
>
> Your observation is spot on. We will emphasize Lines 138-139 to highlight that when the  domain can be non-Markovian or has partial observability, the curse of horizon might be provably unavoidable in general (see Thm 2 in the work by Jiang et al. [46]).
>
> > “the main Theorem 2 and 3 rely on knowing the confidence interval defined below Line 161”
>
> Thank you for bringing this up. We now see that our phrasing of sentences between Lines 151-161 can confuse the reader. We will update this paragraph to highlight that **no oracle knowledge is required** to compute the confidence intervals (CI). Instead, these CI are computed from the off-policy data by leveraging the insight developed in Line 152 and combining it with the CI proposed by Thomas et al. [92] for bounding the mean.
>
> > “It is only universal under very simple settings and has very limited usage.”
>
> To avoid confusion for future readers we will update the contributions section in Line 56 to emphasize that UnO does not require a simplified setting like discrete distributions or oracle knowledge of CIs, provides a universal method to estimate and bound any parameter of the return distribution in the off-policy setting, and can be useful for a wide range of scenarios (Markovian, non-Markovian, fully observable, partially observable, and discrete and smooth distribution shifts).

---

### Official Review · Reviewer_2SVq · 2021-07-15

**Rating:** 7
**Confidence:** 3

**Summary:**

This paper proposes UnO, which is an off-policy evaluation method that estimates any distributional parameter (mean, variance, etc.) simultaneously by estimating the entire distribution of returns and it provides high-confidence bounds for these parameters. Moreover, UnO can be used in settings with partial-observability, non-Markovianity, non-stationarity, and distribution shifts.

**Limitations And Societal Impact:**

Authors adequately addressed the limitations and potential negative societal impact of their work.

**Main Review:**

The paper is very easy to follow despite the fact that it presents a lot of ideas, I particularly liked the visualization in Figure 3. Although there is a lot of content in the paper, I think everything was explored with sufficient depth. Only questions I had about the technical parts were about how to address cases with unknown $\beta$'s or unknown $\epsilon$'s and these limitations are discussed in Appendix B.2. Maybe, more of Appendix B.2 can be incorporated into the main paper in the final version using the additional page.

As far as I understood, the main advantage of UnO is to obtain estimates for a variety of useful parameters (with high-confidence bounds that all hold simultaneously) without relying on individual methods to estimate each parameter separately. Then, for UnO to be a feasible alternative to such individual methods, it is crucial for it to be at least competitive with respect to state-of-the-art solutions. However, experiments are lacking in terms of comparisons with the existing literature. Only comparison that is made is with respect to [92] and only in terms of estimating mean and variance. I would like to see more thorough comparisons with the existing methods in terms of all parameters considered in Figure 4.

Again as far as I understood, the reason why comparisons are lacking is that there is no method that can estimate other parameters in confounded and non-Markovian environments. I am not too familiar with the literature on off-policy evaluation but if that is the case, then it should be highlighted in the introduction that UnO is the first method to estimate these parameters in such environments. Even then, I would still like to see comparisons that evaluate whether UnO is in fact able to learn better estimates in confounded and non-Markovian environments (compared with methods that ignore confoundedness and non-Markovianity).

**Time Spent Reviewing:**

4

---

> ### Author Response · Authors · 2021-08-10
> **Response to reviewer 2Svq**
>
> Thank you for your support!
>
> > “Maybe, more of Appendix B.2 can be incorporated into the main paper in the final version using the additional page.”
>
> That’s a great idea. We will try to do this if an additional page is allowed beyond the 9 already provided.
>
> > “it should be highlighted in the introduction that UnO is the first method to estimate these parameters in such environments”
>
> Thank you for the suggestion, we will try to emphasize this under the contributions on Line 56.
>
> >”evaluate whether UnO is in fact able to learn better estimates in confounded and non-Markovian environments (compared with methods that ignore confoundedness and non-Markovianity)”
>
> To the best of our knowledge, there are no prior works that estimate or bound CVaR or quantiles of the return distribution in the off-policy setting, even if confoundedness and non-Markovianity is ignored. Nearly all of the literature focuses on estimating and bounding the expected return (mean), with [92] being the exception that focused on variance. We will also emphasize this under the contributions on Line 56.

---

### Official Review · Reviewer_ZS4a · 2021-07-16

**Rating:** 7
**Confidence:** 4

**Summary:**

The paper introduces UnO, a framework for estimating confidence bounds for many parameters of the reward in an off-policy setting and with finite state-action spaces. In this work, first an estimator for the CDF of the return is provided, which makes use of an importance sampling correction. From this estimator, it is easy to derive estimators for many useful parameters. Although these estimators are in general biased, it is possible to derive high confidence bounds for any of these parameters.

**Limitations And Societal Impact:**

Yes

**Main Review:**



The paper is very well written and easy to follow. The authors provided a very extensive literature review on off-policy parameter estimation methods.

One thing that is missing in the paper is an analysis of which assumptions are needed in order to apply a concentration inequality for such parameters. Since the proposed estimators rely on importance sampling, a lot of concentration inequalities are not suitable. For instance, standard high confidence bounds require some moments of the importance weights to be bounded, which reflects on a condition on the Renyi divergence between the target and behavioral policies[1]. It is not clear to me how these conditions will change when computing a bound for a general nonlinear function of F. In general, these concentration inequalities might be impossible to apply without further analysis.

I am happy to increase my score if the authors will clarify in the paper the issue above.

[1]Alberto Maria Metelli, Matteo Papini, Francesco Faccio, and Marcello Restelli. Policy optimization via importance sampling. In Advances in Neural Information Processing Systems, pp. 5442–5454, 2018.

**Time Spent Reviewing:**

5

---

> ### Author Response · Authors · 2021-08-10
> **Response to reviewer ZS4a**
>
> Thank you for your careful questions and bringing up the interesting work by Metelli et al. [2018].
>
> >”estimating confidence bounds for many parameters of the reward in an off-policy setting and with finite state-action spaces”
>
> Thank you for mentioning this. We will move the footnote from page 3 to the notation section to emphasize that we only consider finite states and actions for notational simplicity, and that  the results extend directly to the continuous MDPs. In Fig 4, we also have experimental results for the diabetes treatment domains and the POMDP domain, which both have continuous states. Further discussion on this topic is available in the response to Reviewer 4’s feedback.
>
> > “which assumptions are needed in order to apply a concentration inequality”
>
> Regarding the moment of importance weights, Assumption 1 (Line 84) is sufficient for constructing confidence intervals as it ensures that the denominator in the IS ratios are bounded below by $\varepsilon$ (it is not required that $\varepsilon$ is known). This in turn implies that importance ratios are bounded above. This assumption is common both in the off-policy literature [A,B,C] and in real applications [D]. In terms of the divergence measure used by Metelli et al. [2018], this bound on the importance ratio is related to the exponentiated $\alpha$-Renyi-divergence for $\alpha=\infty$. We will add this discussion along with the discussion of the support assumption on Line 640.
>
> As importance ratios are bounded, the CDF estimate $X$ (Line 152) for a given keypoint $\kappa$ is also a bounded random-variable, and thus all its moments are also bounded. This allows us to leverage concentration inequalities for bounded random variables to obtain a confidence band for the entire CDF using Eqn 5.
>
> > “It is not clear to me how these conditions will change when computing a bound for a general nonlinear function of F”
>
> We now see that we could have better emphasized that for both Theorem 2 and 3, the exact same Assumption 1 is sufficient for bounding any general nonlinear function $\psi(F)$. This is possible because the only thing we need in Eqn 6 to bound $\psi(F)$ is a confidence band on the CDF. And as discussed above, this confidence band can be obtained under Assumption 1.
>
> Thank you for your question, which helps us to see parts of the paper that should be clarified! If you feel there is anything that could still benefit from more clarification, do let us know. We will be happy to elaborate further here and also update the draft to help future readers.
>
> ----
> References
>
> [A] Kallus, Nathan, and Masatoshi Uehara. "Double reinforcement learning for efficient off-Policy evaluation in Markov decision processes." JMLR (2020).
>
> [B] Xie, Tengyang, Yifei Ma, and Yu-Xiang Wang. "Towards optimal off-policy evaluation for reinforcement learning with marginalized importance sampling." arXiv preprint arXiv:1906.03393 (2019).
>
> [C] Yang, Mengjiao, et al. "Off-policy evaluation via the regularized Lagrangian." arXiv preprint arXiv:2007.03438 (2020).
>
> [D] Theocharous, Georgios, Philip S. Thomas, and Mohammad Ghavamzadeh. "Personalized ad recommendation systems for life-time value optimization with guarantees." Twenty-Fourth International Joint Conference on Artificial Intelligence. 2015.

---

> > ### Comment · Reviewer_ZS4a · 2021-08-25
> > **Regarding Assumption 1**
> >
> > I thank the authors for clarifying my questions. In the univariate gaussian case, I believe that Assumption 1 is equivalent to assuming that the variance of the target policy is lower than the variance of the behavioral policy, which might be inconvenient when one wants to estimate the performance (or some statistics) of a more exploratory policy, before eventually using it in the environment. When estimators based on Assumption 1 are used in an iterative process, this might be an issue because the target policy tends to become deterministic very fast.
> >
> > Now, if Assumption 1 was relaxed, of course, we would lose all the universality of this method. Can there still be some guarantees by assuming only that some of the $\alpha$-moments of the importance weight are bounded? I believe that when using the weighted importance sampling estimator Assumption 1 could be relaxed.

---

> > > ### Author Response · Authors · 2021-08-26
> > > **Assumption 1 can be relaxed**
> > >
> > > Thanks for pointing out this nuance that we should discuss further in the supplementary materials. We will add further discussion of the various collections of assumptions that provide different guarantees related to valid confidence intervals and consistency, with an emphasis on prior works that analyze bounds on the mean of importance sampling estimators [see Note 1 below] when bounds on the $\alpha$-moments are present [E,F], and other works that enable valid confidence intervals via clipping of importance weights when Assumption 1 is relaxed to only absolute continuity  [see Note 2 below]. Furthermore, prior work has also shown how even the assumption of absolute continuity can in some cases be removed (See discussion around Eqn 8 in the appendix of the work by Thomas et al. [G]).
> > >
> > > Further, as the reviewer pointed out, WIS might also be helpful in relaxing the assumptions on the IS ratios. We will highlight Line 158 to emphasize this nuanced case wherein WIS-based mean bounds [I] can also be used along with the WIS-based UnO estimator (Line 775 in Appendix D) to get a valid confidence band for the entire CDF.
> > >
> > > Apart from the results for confidence intervals, consistency results for our estimator rely upon finite variance, which can be achieved by either assuming that the $\alpha$-divergence is bounded (for $\alpha=2$) or the assumption we have currently (bounded IS ratios).
> > >
> > > However, we plan to still include Assumption 1 for simplicity in the main presentation to avoid all of these nuanced special cases.
> > >
> > > -----
> > > [Note 1] As discussed in Lines 151-158, in our case, the concentration inequalities for the mean are used to compute the bounds on the CDF at key points which are then combined using Theorem 2 to get the confidence band for the entire CDF.
> > >
> > > [Note 2] See the work of Thomas et al. [G (Theorem 1)] for removal of the upper bound on the importance weights when lower-bounding the mean, and the work of Chandak et al. [H (Theorem 5)] for removal of the upper bound on the importance weights when upper-bounding the mean)
> > >
> > > Edit: Typo in [Note 2].
> > >
> > > -----
> > > References
> > >
> > > [E] Metelli, Alberto Maria, et al. "Importance Sampling Techniques for Policy Optimization." J. Mach. Learn. Res. 21 (2020): 141-1.
> > >
> > > [F] Metelli, Alberto Maria, et al. "Policy optimization via importance sampling." arXiv preprint arXiv:1809.06098 (2018).
> > >
> > > [G] Thomas, Philip, Georgios Theocharous, and Mohammad Ghavamzadeh. "High-confidence off-policy evaluation." Proceedings of the AAAI Conference on Artificial Intelligence. Vol. 29. No. 1. 2015.
> > >
> > > [H]  Chandak, Yash, Shiv Shankar, and Philip S. Thomas. "High-Confidence Off-Policy (or Counterfactual) Variance Estimation." arXiv preprint arXiv:2101.09847 (2021).
> > >
> > > [I] Kuzborskij, Ilja, et al. "Confident off-policy evaluation and selection through self-normalized importance weighting." International Conference on Artificial Intelligence and Statistics. PMLR, 2021.

---

### Official Review · Reviewer_wkXw · 2021-07-20

**Rating:** 9
**Confidence:** 3

**Summary:**

The paper proposes an algorithm for estimating any parameter of the distribution of returns in an off-policy manner, and derives confidence intervals for the quantities of interest. The paper also shows how the proposed algorithm can be extended to deal with many issues commonly faced when applying RL to real world problems (nonstationarity, non-markovian observations, etc.).

**Limitations And Societal Impact:**

The paper does a thorough and honest job of addressing limitations and potential negative societal impacts.

**Main Review:**

I strongly recommend accepting the paper for publication.

**Quality:**

Overall the quality of the paper is very high. The methods are derived in a principled fashion, and important properties (guaranteed coverage etc.) are proved. The experiments also seem to be well done and are helpful for illustrating properties of the proposed method.

**Significance:**

The proposed algorithm is very likely to be significant for both practitioners, as well as researchers to build upon. The generality and effectiveness of the proposed algorithm seems extremely useful, as are the confidence bounds that are important for any real-world application of off-policy learning.

**Clarity:**

The paper is written clearly, although somewhat difficult to understand due to the complicated concepts involved and the large amount of ground covered within the paper. The paper is quite dense with notation, but the authors have helpfully included a notation table in the appendices.

**Originality:**

The proposed algorithm is novel to the best of my knowledge.

**Miscellaneous comments, questions, and suggestions to improve the paper:**

Citing authors by last name and date instead of with an arbitrary number makes it possible for the reader to recognize what paper is being referred to if they're already familiar with it. It's also nice to give explicit credit to authors whose work is being built upon.

Line 76: Why use discounting when there's a horizon length $T$? Is it just for the sake of generality?

Line 84: It would be nice to comment on assumptions, explaining why they were made, any ramifications/limitations due to them, whether they are common, restrictive, etc.

Line 273: What is the utility of confidence intervals that don't hold with the specified probability? Are they even confidence intervals in that case?

It might be better to plot confidence intervals rather than standard error. The reason I write this is that non-overlapping confidence intervals imply statistical significance, whereas non-overlapping standard error does not. This doesn't seem to matter as much for this paper though, as the experiments focused more on investigating properties of the proposed algorithm rather than comparing it against competitors (of which there seem to be few).

**Time Spent Reviewing:**

5

---

> ### Author Response · Authors · 2021-08-10
> **Response to Reviewer wkXw**
>
> Thank you for your support!
>
> >”Line 76: Why use discounting when there's a horizon length ? Is it just for the sake of generality?”
>
> Indeed, it is just for generality.
>
>
>
> >”Line 84: It would be nice to comment on assumptions”
>
> Thank you for the suggestion. We will incorporate some of the discussion from Appendix B.2 “Knowledge of Subset Support” into the main paper.
>
>
>
> > “Line 273: What is the utility of confidence intervals that don't hold with the specified probability?”
>
> Bootstrap based intervals are confidence intervals that are typically valid asymptotically (not for finite sample sizes), but which can often provide relatively tight intervals. In comparison, confidence intervals from concentration inequalities (Hoeffding, Bernstein, etc.) hold for any finite sample sizes, but provide relatively loose intervals. To answer your question directly, bootstrap confidence intervals (e.g., those described in Efron and Tibshirani’s popular 1994 book) are useful when the confidence intervals that hold with the specified probability are too loose to be useful.
>
> We provide versions of UnO based on both bootstrap and concentration inequalities (Uno-Boot and UnO-CI) so that an end-user can choose which of these is more suitable for their application.

---

### Decision · Program_Chairs · 2021-09-27

**Decision:**

Accept (Poster)

**Comment:**

The paper studies off-policy evaluation through estimates and high-confidence bounds on any parameter of the return distribution, such as its mean, variance, IQR, etc. The authors make use of importance sampling, an often-used tool in observational studies, proving that it leads to unbiased and consistent estimation of the cumulative distribution of rewards. This is later extended to provide high-confidence bounds on the CDF, and for parameters of the CDF. The implications of various commonly made assumptions are discussed and the estimators are evaluated empirically.

The reviews were predominantly positive, arguing for acceptance with one exception. One of the main limitations raised by reviewer j4b9 was that the return distribution is assumed to be discrete. The authors clarified in their response that this limitation may be removed. Additional clarifications were made by the authors which imply only small modifications to the manuscript.